# SPIQA: A Dataset for Multimodal Question Answering on Scientific Papers

**Shraman Pramanick**[*1,2◇]    **Rama Chellappa**[2]    **Subhashini Venugopalan**[*1†]

[1]Google Research    [2]Johns Hopkins University

https://huggingface.co/datasets/google/spiqa

https://github.com/google/spiqa

## Abstract

Seeking answers to questions within long scientific research articles is a crucial area of study that aids readers in quickly addressing their inquiries. However, existing question-answering (QA) datasets based on scientific papers are limited in scale and focus solely on textual content. We introduce SPIQA (**S**cientific **P**aper **I**mage **Q**uestion **A**nswering), the first large-scale QA dataset specifically designed to interpret complex figures and tables within the context of scientific research articles across various domains of computer science. Leveraging the breadth of expertise and ability of multimodal large language models (MLLMs) to understand figures, we employ automatic and manual curation to create the dataset. We craft an information-seeking task on interleaved images and text that involves multiple images covering plots, charts, tables, schematic diagrams, and result visualizations. SPIQA comprises 270K questions divided into training, validation, and three different evaluation splits. Through extensive experiments with 12 prominent foundational models, we evaluate the ability of current multimodal systems to comprehend the nuanced aspects of research articles. Additionally, we propose a Chain-of-Thought (CoT) evaluation strategy with in-context retrieval that allows fine-grained, step-by-step assessment and improves model performance. We further explore the upper bounds of performance enhancement with additional textual information, highlighting its promising potential for future research and the dataset's impact on revolutionizing how we interact with scientific literature.

## 1 Introduction

Surfacing pertinent information within the context of academic research articles is an essential area of study, as it empowers students and researchers to efficiently address their queries which are naturally triggered when reading a scientific paper. However, existing question-answering (QA) datasets anchored on academic articles are limited in terms of scale [75, 21, 30, 14, 35]. This limitation arises due to the complexity and cost associated with curating questions, as understanding such articles demands domain-specific expertise, a detailed understanding of the topic, and a significant amount of time. Additionally, prior QA datasets in this domain only analyze the abstracts, conclusion [59, 21] and the textual paragraphs [58, 56, 69, 30, 14, 66, 35] of scientific articles, overlooking the wealth of information presented in meticulously crafted figures and tables, and hence, fail to leverage and analyze the rich, multidimensional data embedded in these visual elements, which are crucial for a comprehensive understanding of the research presented.

Numerous datasets have been curated to evaluate the QA abilities of Large Language Models (LLMs) on various documents, including factual documents [85, 62, 22, 86, 32, 12, 68, 70], book chapters [28, 29, 51], news articles [74, 36] and more. However, understanding scientific papers poses unique challenges as the systems must comprehend underlying theoretical implications with domain-specific terminologies and verify claims with experimental results and visualizations. There are also several datasets that focus on the comprehension of standalone science diagrams [27, 24, 18, 65],

---

[*]equal technical contribution, [◇]work done as a student researcher at Google Research.

[†]Corresponding author (vsubhashini@google.com)

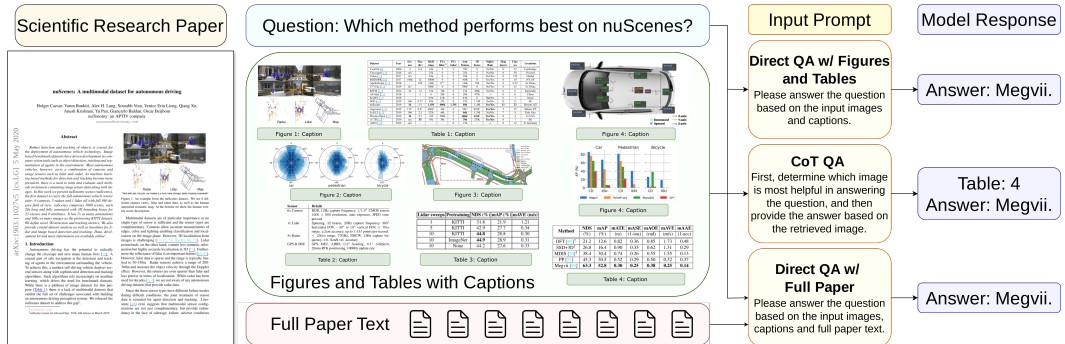

Figure 1: **Illustration of the SPIQA tasks.** Given a question anchored in figures from a research paper, we evaluate the capabilities of multimodal LLMs in comprehending and integrating information across multiple figures, tables and paper text.

mathematical figures [45, 87], charts [5, 48, 49], plots [50], tables [60, 76, 9, 8, 11, 26, 52] and retrieval [71]. However, simultaneously reasoning over all figures and associated text in a scientific article necessitates both multimodal and long-context capabilities.

In contemporary research, figures and tables with associated captions are crucial to understand the motivation and contributions of a work. Comprehending such multimodal content presents a significant real-world challenge. In this work, we introduce SPIQA (**S**cientific **P**aper **I**mage **Q**uestion **A**nswering), the first large-scale QA dataset specifically designed to interpret complex figures and tables in the context of scientific research articles across various domains of computer science. We also develop three interleaved image-text tasks using the SPIQA dataset to assess growing long-context capabilities of MLLMs: (1) Direct QA with figures and tables, where the systems require to answer questions after seeing all figures and tables from a scientific paper, (2) Direct QA with full paper, where the systems analyze the whole paper text along with figures and tables to answer questions and (3) CoT QA, a chain of thought (CoT) retrieval based QA where systems need to first identify helpful figures and then answer the question, allowing to evaluate the models for fine-grained reasoning and grounding capability. We collect the PDFs and source TeXs of 26K papers from various domains of computer science published in top-tier academic conferences and generate 270K question-answer-rationale triplets focusing on the papers' figures and tables. We further filter and augment two existing scientific QA datasets, QASA [35] and QASPER [14], to identify questions requiring reasoning over figures, tables, and textual paragraphs. Overall, SPIQA contains one training, one validation, and three evaluations split with varying difficulty levels, allowing us to fine-tune and assess the capabilities of large multimodal systems to understand complex scientific research papers. An illustration of the three proposed tasks with example prompts is presented in Figure 1.

We conduct extensive experiments on the SPIQA datasets evaluating the comprehension abilities of several closed large multimodal models, and state-of-the-art open-source models, including Gemini [63, 72], GPT4 [54, 1], Claude-3 [2], LLaVA 1.5 [40], InstructBLIP [13], XGen-MM [64], InternLM-XC [15], SPHINX-v2 [17] and CogVLM [80]. We further fine-tune InstructBLIP and LLaVA 1.5 on the SPIQA training set and observe significant improvement compared to zero-shot evaluation, indicating potential pathways for designing specialized systems for scientific QA in the future. In addition to reporting performances on traditional QA metrics, we introduce a novel LLM-based evaluation metric, LLMLogScore (L3Score), which incorporates the confidence of LLMs for assessing the equivalence of answers with the ground-truths based on the log-likelihood token probabilities. We demonstrate the effectiveness of L3Score for automated free-form QA evaluation over existing LLM-based scores [47, 91, 34] which use sensitive rating scales.

In summary, our contributions are: $(i)$ We curate SPIQA, the first large-scale QA dataset specifically designed to interpret complex figures and tables in the context of scientific research papers across various computer science domains. $(ii)$ We develop three well-designed tasks for direct and retrieval-based CoT question answering to access baseline systems' step-by-step fine-grained reasoning capabilities. $(iii)$ We propose LLMLogScore (L3Score), a novel LLM-based evaluation metric for free-form QA. L3Score incorporates the confidence of LLMs to evaluate the quality of candidate answers using log-likelihood token probabilities. $(iv)$ We perform extensive experiments to demon-

strate the value of SPIQA and LLMLogScore to assess multimodal and long-context capabilities of several closed and open-sourced MLLMs.

## 2 Related Works

| Dataset | Free-form Questions? | Question Generation | Num QA | Num Abstracts / Papers | Paper Source | Domain | Questions based on | |
|---|---|---|---|---|---|---|---|---|
| | | | | | | | Full Text | Figs & Tabs |
| BioAsq [75, 30] | ✓ | Human experts | 3.2K | – | PubMed | Biomedical | ✗ | ✗ |
| BioRead [58] | ✗ | Cloze-style | 16.4M | 3.4M papers | PubMed | Biomedical | ✓ | ✗ |
| BioMRC [59] | ✗ | Cloze-style | 812K | 25M abstracts | Pubtator | Biomedical | ✗ | ✗ |
| emrQA [56] | ✗ | Cloze-style | 455K | 2.4K clinical notes | i2b2 datasets | EMRs | ✓ | ✗ |
| MedHop [82] | ✗ | Cloze-style | 2.5K | 24M abstracts | Medline 2016 | Molecular Biology | ✗ | ✗ |
| PubMedQA [21] | ✓ | Human experts | 1K | 120K abstracts | PubMed | Biomedical | ✗ | ✗ |
| BioASQ-QA [31] | ✓ | Human experts | 4.7K | – | PubMed | Biomedical | ✗ | ✗ |
| CliCR [69] | ✗ | Cloze-style | 105K | 12K clinical reports | BMJ Case Reports | Medical | ✗ | ✗ |
| ArgSciChat [66] | ✓ | Human Experts | 41 dialogues | 20 papers | arXiv | NLP | ✓ | ✗ |
| QASPER [14] | ✓ | Human experts | 5K | 1.5K papers | S2ORC | NLP | ✗ | ✗ |
| QASA [35] | ✓ | Human experts | 1.8K | 112 papers | S2ORC | AI/ML | ✓ | ✗ |
| **SPIQA(Ours)** | ✓ | LLMs + Human experts | 270K | 25.5K papers | arXiv | Computer Science (all) | ✓ | ✓ |

Table 1: **Comparison of SPIQA with existing scientific question answering datasets.** SPIQA is a large-scale free-form QA corpus that focuses on the holistic aspects of scientific papers, including figures, tables, and full paper text. A dash (−) indicates the number is unreported.

Question-answering on long documents is a challenging real-world task that has attracted increasing attention in recent years, following the success of the long-context reasoning ability of LLMs [41, 10, 72, 63, 54, 2, 37, 73, 16]. Though there exist numerous general-domain document QA datasets [85, 62, 22, 86, 32, 12, 68, 70, 79, 61, 84, 89, 78, 74, 36], understanding scientific papers requires domain-specific expertise and reasoning capability, and poses a more significant challenge.

**Datasets for QA on Scientific Papers.** In the early days, cloze-style academic paper QA datasets [58, 56, 83, 69, 59] were automatically constructed by extracting entities and relations and matching them with structure knowledge resources. The questions in such datasets follow a pre-defined format; hence, they are unsuitable for real-world usage where the reader asks detailed open-ended questions [32]. To overcome these issues, PubMedQA [21], BioAsq [30] and QASPER [14] construct corpora of 1K, 3.2K, and 5K human-written questions, respectively. However, the annotators of these datasets only read the abstracts when writing the questions, and hence, most questions are simple and can be answerable in yes/no format or with short extractive spans. ArgSciChat [66] proposes a dataset of argumentive dialogues between scientists on 20 NLP papers. Closer to our work, QASA [35] generates 1798 free-form advanced questions on AI/ML papers where the annotators read the whole paper. However, QASA questions are answerable just from the text paragraphs, neglecting the structured information present in terms of figures and tables. Table 1 presents detailed comparisons.

**Datasets for QA on Scientific Diagrams.** Solving mathematical problems in a visual context has emerged as a complex reasoning task for MLLMs. Prior attempts, such as GeoQA [7], UniGeo [6], and Geometry3K [46], have exclusively focused on solving geometry-oriented questions. In a different line of research, datasets like DVQA [23], LEAF-QA [5], LEAFQA++ [67], FigureQA [24], PlotQA [50], and ChartQA [48] have been constructed to solve plot and chart-oriented questions. Additionally, there are datasets for QA purely on tabular data, including WTQ [60], TableQA [76], SQA [20], HiTab [11], AIT-QA [26], FetaQA [52], MultiTabQA [55]. More recently, MathVista [45], MathVerse [87], and ArXivQA [38] have integrated different scientific diagrams to develop benchmarks with a wider variety of tasks. However, existing datasets focus on asking questions about standalone figures or tables (additional comparisons are in Appendix Sec. C). SPIQA bridges this gap by proposing a new scientific QA benchmark that includes questions that require simultaneous reasoning over figures, tables, and textual paragraphs, allowing a more integrated understanding of scientific documents.

Our proposed SPIQA dataset distinguishes itself from existing scientific question answering (QA) benchmarks in several significant ways. Firstly, it presents a large-scale, open-ended QA dataset to encompass diverse domains of computer science, providing a comprehensive evaluation framework for scientific QA systems. Secondly, the questions and answers in SPIQA necessitate an understanding of complex visual elements, including figures and tables, in conjunction with textual content and domain knowledge. This requirement simulates real-world scenarios where researchers must synthesize information from multiple sources to answer complex questions. Thirdly, we introduce a new Chain-of-Thought (CoT) QA paradigm in interleaved image-text comprehension, which involves a two-stage process wherein models first identify relevant figures and tables, and then generate answers based on

this context. This step-by-step approach enables a more fine-grained assessment of the reasoning capabilities of baseline systems, providing valuable insights into their strengths and limitations.

## 3 SPIQA Dataset and Tasks

### 3.1 Collection Guidelines and Task Formulation

Existing scientific QA benchmarks [58, 69, 56, 21, 30, 14, 66, 35] primarily focus only on text from the main body of the paper, overlooking the wealth of information presented as figures and tables. Further, curating QA anchored in research articles requires domain expertise and a detailed understanding of the paper, making annotations expensive and resulting in smaller-scale data. Our dataset, SPIQA, bridges this gap by systematically curating a large-scale QA benchmark focusing on every aspect of scientific research documents - main text, figures, tables, and their captions, and thus pushing AI systems towards a robust understanding of research articles. Moreover, we annotate which figures and tables help answer a question to evaluate the grounding and CoT reasoning capabilities of large multimodal systems.

While collecting and annotating the SPIQA benchmark, we adhere to the following collection guidelines: $(i)$ We identify 19 different top-tier academic research conferences covering a wide variety of computer science domains where the papers are licensed permissively. $(ii)$ We collect 27K PDFs and corresponding TeX sources of research papers presented in those conferences between 2018-2023. Using peer-reviewed articles helps us to maintain the quality of SPIQA. The TeX sources provide high-resolution figures and table TeXs, which can not directly be extracted from the PDFs. $(iii)$ We curate questions of varying levels of difficulty based on the collected papers, requiring robust long-context understanding of different kinds of figures and tables associated with their captions and the main body of the paper. $(iv)$ Lastly, we identify subsets of questions in two existing datasets, QASA [35] and QASPER [14], which require understanding of figures and tables with the main text, and treat them as two additional evaluation splits of SPIQA. We formulate three novel tasks for the comprehensive evaluation of various QA systems on SPIQA:

- **Direct QA with Figures and Tables**: In this task setup, we provide all figures and tables with their captions from a paper. The systems are then required to answer questions that necessitate reasoning with one or more figures and tables.

- **Direct QA with Full Paper**: Here, we provide the entire paper, including the main text, figures, and tables with captions, and ask the systems to answer questions. While the paper text helps by providing additional information, the systems require strong long-context understanding capability to reason over full text.

- **CoT QA**: To assess the step-by-step reasoning abilities of MLLMs, CoT QA requires systems first to identify relevant figures and tables and then answer the question. For simplicity, CoT QA does not involve full paper text. Detailed prompts for each task are included in the code.

### 3.2 SPIQA: Data Collection, Question Generation and Filtering

**Collection of Paper PDFs and TeX Sources.** We begin by collecting a large corpus of scientific research papers across all domains of computer science, focusing on open-source publications. We identified 19 top-tier conferences, as detailed in Figure 2, and gathered the paper PDFs published at these venues between 2018 and 2023. We then filtered this collection to include papers whose TeX sources could be downloaded using the python arXiv API[1]. This process resulted in 25,859 peer-reviewed papers with corresponding TeX sources, which provide high-resolution figures, table TeXs, and the main body of the articles. We additionally use PDFFigures 2.0[2] to crop figures from the paper to account for figures that are missed when processing the TeX sources. Overall, we gather 152,487 figures, 117,707 tables, and their captions from 25,859 papers with corresponding main text. It is worth noting that due to our structured step-by-step paper collection strategy, we will have the opportunity to expand SPIQA in the future with papers published in later years and in other open-sourced research fields. Table 2 shows a detailed classification of the collected figures in granular subcategories.

**Automatic Question Generation and Filtering.** After gathering the main text, figures, and tables from research papers, the next step is to generate high-quality question-answer pairs that cover all

---

[1]python arXiv API: https://github.com/lukasschwab/arxiv.py
[2]PDFFigures 2.0: https://github.com/allenai/pdffigures2

| Statistics | Numbers |
|---|---|
| Total papers | 25,859 |
| Published between | 2018 - 2023 |
| Total tables | 117,707 |
| Total figures | 152,487 |
| Figure subcategories | |
| - Schematics | 45396 |
| - Plots and charts | 72327 |
| - Visualizations | 28103 |
| - Others | 6661 |
| Total generated QAs | 270194 |
| Maximum question length | 194 |
| Maximum answer length | 333 |
| Average question length | 12.98 |
| Average answer length | 14.56 |

Table 2: **Statistics** of the collected research papers and generated questions.

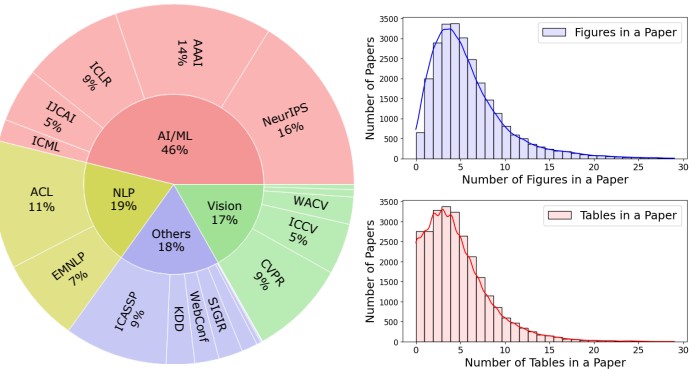

Figure 2: **Source of the collected papers and distribution of figures and tables per paper.** We collect papers from 19 top-tier conferences in computer science published between 2018 and 2023.

aspects of the articles. Manually annotating quality questions requires domain expertise and a deep understanding of the research papers, resulting in existing human-annotated datasets being small-scale or solely based on abstracts [75, 21, 30, 14, 35]. To bridge this gap, we automatically generate QAs by leveraging recent advances in powerful multi-modal large language models. We first conducted a pilot study by selecting 30 papers from various domains and experimenting with multiple models. In our approach, we presented one figure or table to the model, along with passages referencing the figure. We then asked the models to generate a question, answer, and rationale that requires a holistic understanding of the figure or table in the context of the paper. We manually verified the quality of the generated QAs by each model using the following criterion: $(i)$ Answering the question would require understanding of the figure or table. $(ii)$ The generated answer is correct and to the point. $(iii)$ The question is neither too trivial nor too specific to the figure or table. The questions looked promising and we proceeded to generate 270,194 QAs over 25,859 papers using Gemini 1.5 pro.

After generating QAs, we divide the dataset into three splits: 200 papers for evaluation, 200 for validation, and the remaining 25,459 for training, ensuring each split consists of papers from all domains. Despite the good question quality, we perform additional manual filtering on the evaluation set to exclude any inconsequential and incorrect QAs. In addition to the three criteria of the pilot study, the filtering guideline contains the following standards: $(iv)$ If two or more questions from a paper are similar, keep one. $(v)$ If the question is entirely based on the passage, discard the QA. $(vi)$ If the answer is not clear, e.g., the answer says '*It is hard to answer the question based on the given information*' or '*The answer is not evident from the given passage*', discard the QA. $(vii)$ If the QA includes phrases like '*Based on the passage,*' modify it because we show all figures and tables to the model at once during evaluation. Appendix Sec. G includes the QA generation prompts, filtering guidelines and UI. We double annotated a small set of questions and found 88% agreement between annotators on which questions to discard. Overall, only around 11% of questions were discarded when filtering the evaluation set, and hence no additional filtering was performed on the train and validation splits. After filtering, the final evaluation set contains 666 questions, and we refer to this split as test-A.

**Additional Evaluation Sets with Human Written QAs.** Since the questions and answers in test-A are automatically generated by Gemini, to develop a robust and complete QA benchmark on scientific articles, we curate two additional test sets with human-written QAs by utilizing two existing datasets, QASA [35] and QASPER [14]. However, the human annotators for these datasets mainly considered the paper text. We aim to identify the questions in these datasets that require reasoning with figures or tables for comprehensive answers. For every question, both QASA and QASPER provide evidence sentences from the paper, which are crucial for answering. We parsed these evidence sentences to look for phrases like '*According to Figure x,*' '*As shown in Table y,*' '*Figure z presents,*' and separate such questions, and treat the corresponding figure and tables as evidence. We collect the PDFs for the papers in QASA and QASPER from arXiv, and extract the figures using PDFFigures 2.0. We conduct additional manual filtering to verify that the identified evidence figures help answer the questions. This process resulted in 228 questions from QASA and 493 questions from QASPER, where figures or tables are helpful for answering. We refer to these sets generated from QASA and QASPER as test-B and test-C, respectively. We additionally took a mix of human and model generated questions, and asked annotators to indicate whether each question was generated by a

| Method | SPIQA test-A | | | | | SPIQA test-B | | | | | SPIQA test-C | | | | |
|---|---|---|---|---|---|---|---|---|---|---|---|---|---|---|---|
| | M | R-L | C | B-F1 | L3S | M | R-L | C | B-F1 | L3S | M | R-L | C | B-F1 | L3S |
| *Zero-shot Closed-Weight MLLMs* | | | | | | | | | | | | | | | |
| Gemini Pro Vision [72] | 22.9 | 38.3 | 124.6 | 64.87 | 43.85 | 9.9 | 19.0 | 29.1 | 54.83 | 31.84 | 11.6 | 19.4 | 47.8 | 48.95 | 31.98 |
| Gemini 1.5 Flash [63] | 25.4 | 38.8 | 110.9 | 65.84 | 54.20 | 11.5 | 19.4 | 24.4 | 56.32 | 36.04 | 14.4 | 18.1 | 45.5 | 48.79 | 36.67 |
| Gemini 1.5 Pro [63] | 23.4 | 35.5 | 87.1 | 64.36 | 53.49 | 10.8 | 19.3 | 26.8 | 56.62 | 43.27 | 12.6 | 16.8 | 40.2 | 47.51 | 36.72 |
| Claude 3 (Opus) [2] | 25.0 | 41.5 | 120.2 | 65.84 | 61.26 | 12.7 | 19.2 | 17.0 | 57.03 | 49.54 | 15.5 | 29.7 | 92.6 | 52.35 | 43.88 |
| GPT-4 Vision [1] | 23.1 | 37.7 | 113.8 | 64.01 | 56.67 | 12.2 | 18.8 | 23.7 | 55.09 | 43.62 | 15.2 | 22.9 | 75.5 | 51.02 | 40.85 |
| GPT-4o [54] | 25.5 | 42.2 | 133.7 | 66.14 | 64.00 | 10.7 | 18.9 | 31.8 | 53.73 | 46.22 | 15.6 | 31.3 | 98.4 | 53.57 | 46.68 |
| *Zero-shot Open-Weight MLLMs* | | | | | | | | | | | | | | | |
| SPHINX-v2 [17] | 2.6 | 3.2 | 13.4 | 6.25 | 3.34 | 0.1 | 0.3 | 0.4 | 2.08 | 1.65 | 1.0 | 3.3 | 11.0 | 8.03 | 3.32 |
| InstructBLIP-7B [13] | 9.5 | 18.9 | 62.6 | 47.70 | 7.50 | 3.5 | 9.5 | 16.3 | 39.62 | 7.07 | 2.8 | 15.5 | 36.6 | 48.45 | 8.79 |
| LLaVA-1.5-7B [40] | 22.6 | 34.7 | 117.8 | 61.61 | 13.86 | 7.7 | 15.5 | 16.8 | 47.21 | 9.63 | 7.0 | 15.1 | 26.7 | 45.55 | 9.53 |
| XGen-MM [64] | 17.3 | 30.6 | 127.0 | 58.41 | 13.74 | 4.4 | 8.0 | 11.1 | 35.49 | 8.18 | 4.2 | 17.4 | 46.4 | 45.25 | 10.66 |
| InternLM-XC [15] | 22.2 | 29.2 | 73.7 | 53.57 | 18.28 | 8.1 | 12.9 | 16.8 | 36.00 | 12.47 | 8.5 | 11.4 | 20.5 | 34.58 | 11.84 |
| CogVLM [80] | 20.4 | 27.9 | 59.2 | 51.24 | 16.89 | 7.9 | 16.0 | 26.2 | 43.93 | 9.60 | 9.7 | 13.9 | 24.4 | 42.90 | 12.52 |
| *Fine-tuned MLLMs* | | | | | | | | | | | | | | | |
| InstructBLIP-7B [13] | 17.8 | 32.5 | 110.0 | 62.10 | 43.90 | 8.8 | 17.2 | 28.6 | 52.79 | 31.82 | 10.1 | 22.8 | 69.8 | 50.22 | 33.48 |
| $\Delta$ **InstructBLIP-7B FT - ZS** | 8.3↑ | 13.6↑ | 47.4↑ | 14.40↑ | 36.40↑ | 5.3↑ | 7.7↑ | 12.3↑ | 13.17↑ | 24.75↑ | 7.3↑ | 7.3↑ | 33.2↑ | 1.77↑ | 24.69↑ |
| LLaVA-1.5-7B [40] | 23.8 | 36.0 | 121.2 | 63.74 | 45.45 | 11.0 | 18.4 | 29.5 | 53.13 | 33.50 | 10.5 | 24.1 | 69.6 | 50.15 | 32.40 |
| $\Delta$ **LLaVA-1.5-7B FT - ZS** | 1.2↑ | 1.3↑ | 3.4↑ | 2.13↑ | 31.59↑ | 3.3↑ | 3.1↑ | 12.7↑ | 5.92↑ | 23.87↑ | 3.5↑ | 9.0↑ | 42.9↑ | 4.60↑ | 22.87↑ |

Table 3: **Performance of zero-shot and fine-tuned systems on direct QA with figures and tables.** GPT-4o achieves the state-of-the-art results on test-A and test-C, while Claude-3 performs similarly well as GPT-4o on test-B. M: METEOR, R-L: ROUGE-L, C: CIDEr, B-F1: BERTScore F1 and L3S: L3Score. $\Delta$ shows improvements after fine-tuning.

human, machine or if they're unsure and 85% of the time, annotators indicated they were unsure, indicating the indistinguishability of LLM generated questions from human written ones.

## 3.3 Dataset Analysis

**Splits & Statistics.** The main statistics of the collected papers, generated questions, and data splits are presented in Tables 2 and 4. SPIQA encompasses 25,859 computer science papers published in top-tier conferences between 2018-2023, containing 152,487 figures and 117,707 tables. Figure 2 shows the distribution of figures and tables in every paper. We generate a total of 270,194 QAs focusing on the figures and tables along with the main text of the papers. The questions and answers, on average, contain 12.98 and 14.56 words, respectively. Notably, we observe high variances in their lengths - 20.47 for questions and 243.29 for answers - indicating a diverse range of patterns in SPIQA. The training, validation, and test-A splits include papers from every source conference and ensure that questions from the same paper remain in the same split. The test-B and test-C splits are generated from QASA [35] and QASPER [14] and contain human-written QAs. Examples from the dataset, illustrated in Figure 4, highlight two different questions centered on different types of figures. Appendix Sec. C provides a detailed comparison of SPIQA with existing scientific QA datasets.

**Granularity.** We divide the collected figures in four broad categories - schematic diagrams, plots & charts, visualizations and others. Table 4 presents number of figures in each category across all splits. Table 6 reports performance of baseline models on all types of figure and tables from test-A.

| Split | # Papers | # Ques. | # Figures | | | | # Tables |
|---|---|---|---|---|---|---|---|
| | | | Sche. | Plots & Charts | Vis. | Others. | |
| Train | 25,459 | 262,524 | 44,008 | 70,041 | 27,297 | 6,450 | 114,728 |
| Val | 200 | 2,085 | 360 | 582 | 173 | 55 | 915 |
| test-A | 118 | 666 | 154 | 301 | 131 | 95 | 434 |
| test-B | 65 | 228 | 147 | 156 | 133 | 17 | 341 |
| test-C | 314 | 493 | 415 | 404 | 26 | 66 | 1332 |

Table 4: **Split Statistics of SPIQA.** Train, val, test-A contains LLM-generated QAs; test-B and test-C have human-written QAs. We report the number of tables and figures in each split.

## 4  LLMLogScore (L3Score): An Improved Metric for Free-form QA

Evaluating free-form QA is challenging because the answers are often descriptive and lack a predefined format. Current LLMs generate varied and detailed responses that may appear different but are still accurate, which traditional QA evaluation metrics such as BLEU [57] and ROUGE [39] often fail to capture. Inspired by the ability of LLMs to evaluate natural language generation (NLG) [90, 42, 25], three recent approaches, LAVE [47], LIMA [91], and Prometheus-Vision [34], utilize the in-context capability of instruction-tuned LLMs to rate candidate answers on 3, 6, and 5-point scales, respectively. However, these metrics are highly sensitive to the chosen scale range and do not consider the LLM's confidence in the provided ratings.

| Method | Ret. Acc. | SPIQA test-A QA M | R-L | C | B-F1 | L3S | Ret. Acc. | SPIQA test-B QA M | R-L | C | B-F1 | L3S | Ret. Acc. | SPIQA test-C QA M | R-L | C | B-F1 | L3S |
|---|---|---|---|---|---|---|---|---|---|---|---|---|---|---|---|---|---|---|
| Gemini 1.5 Flash [63] | – | 25.4 | 38.8 | 110.9 | 65.84 | 54.20 | – | 11.5 | 19.4 | 24.4 | 56.32 | 36.04 | – | 14.4 | 18.1 | 45.5 | 48.79 | 36.67 |
| w/ Full Paper | – | 27.1 | 41.5 | 125.2 | 69.20 | 58.12 | – | 13.7 | 24.1 | 53.4 | 59.95 | 37.42 | – | 14.8 | 18.9 | 52.0 | 49.77 | 37.25 |
| w/ CoT | 86.18 | 26.0 | 39.6 | 120.5 | 68.11 | 56.57 | 57.45 | 10.9 | 19.5 | 26.5 | 57.75 | 35.75 | 69.37 | 13.1 | 18.5 | 47.8 | 49.36 | 34.28 |
| Gemini 1.5 Pro [63] | – | 23.4 | 35.5 | 87.1 | 64.36 | 53.49 | – | 10.8 | 19.3 | 26.8 | 56.62 | 43.27 | – | 12.6 | 16.8 | 40.2 | 47.51 | 36.72 |
| w/ Full Paper | – | 27.0 | 40.4 | 116.8 | 69.05 | 61.80 | – | 13.5 | 22.3 | 34.8 | 59.18 | 47.51 | – | 13.2 | 17.5 | 54.3 | 49.25 | 39.48 |
| w/ CoT | 85.88 | 25.6 | 38.7 | 99.5 | 67.15 | 64.68 | 62.28 | 11.2 | 18.6 | 27.0 | 56.46 | 47.38 | 70.79 | 14.6 | 18.7 | 55.7 | 49.79 | 41.12 |
| GPT-4 Vision [1] | – | 23.1 | 37.7 | 113.8 | 64.01 | 56.67 | – | 12.2 | 18.8 | 23.7 | 55.09 | 43.62 | – | 15.2 | 22.9 | 75.5 | 51.02 | 40.85 |
| w/ Full Paper | – | 27.0 | 39.5 | 128.7 | 67.24 | 62.45 | – | 14.8 | 22.1 | 28.3 | 58.33 | 46.63 | – | 15.6 | 23.8 | 94.8 | 52.46 | 42.28 |
| w/ CoT | 83.25 | 25.6 | 38.8 | 94.6 | 66.92 | 63.37 | 60.45 | 11.6 | 20.0 | 27.4 | 57.82 | 45.35 | 66.73 | 14.5 | 19.4 | 56.5 | 50.43 | 43.83 |
| GPT-4o [54] | – | 25.5 | 42.2 | 133.7 | 66.14 | 64.00 | – | 10.7 | 18.9 | 31.8 | 53.73 | 46.22 | – | 15.6 | 31.3 | 98.4 | 53.57 | 46.68 |
| w/ Full Paper | – | 27.4 | 45.2 | 137.5 | 68.75 | 65.89 | – | 14.6 | 24.1 | 55.0 | 59.39 | 47.61 | – | 16.3 | 34.0 | 107.5 | 54.15 | 48.10 |
| w/ CoT | 85.58 | 27.2 | 43.6 | 131.0 | 69.34 | 66.09 | 63.63 | 10.8 | 20.0 | 35.8 | 57.75 | 46.52 | 70.38 | 15.7 | 22.8 | 64.9 | 52.52 | 48.62 |

Table 5: **Performance on direct QA with full paper and CoT QA.** Both tasks help to improve the performance of Gemini and GPT4 models over direct QA with figures and tables. Acc: top-1 accuracy, M: METEOR, R-L: ROUGE-L, C: CIDEr, B-F1: BERTScore F1 and L3S: L3Score.

We alleviate the necessity of a predefined scale range and detailed interpretation of every point in the scale by proposing LLMLogScore (L3Score), which directly uses the log-likelihood probabilities generated by an LLM for evaluating the answers. We use GPT-4o [54] as our LLM, show the model the candidate and the ground-truth answers, and ask if the semantic meaning of the candidate and the ground-truth are similar in the context of the question. The LLM is expected to answer yes or no in a single word. The prompt used for calculating L3Score is in Appendix Sec. F. Next, instead of considering the final response by the LLM, we look into the top-5[3] log probabilities and corresponding tokens, and we define the L3Score metric as follows:

$$\text{L3Score} = \text{softmax}(x)_{yes} = \frac{exp(l_{yes})}{exp(l_{yes}) + exp(l_{no})} \tag{1}$$

$exp()$ is the exponential and $\text{softmax}()$ is the softmax operation, $l_{yes}$ and $l_{no}$ are the log probability for the tokens 'yes' and 'no' and $x$ represents the vector $[l_{yes}, l_{no}]$. Essentially L3Score renormalizes the probabilities of tokens 'yes' and 'no' to sum to 1. However, due to the caveat that we use a closed model, only the top-5 log probabilities are available to us, hence we may need to approximate the log probability if one or both tokens are missing from the top-5. We do it as follows

1. If neither $l_{yes}$ or $l_{no}$ is in the top-5, L3Score $= 0$.
2. If only one of $l_{yes}$ or $l_{no}$ is present, then we approximate the $l_{x_c}$ for the complementary missing token (denoted $x_c$), by considering the minimum (*min*) of the token with the lowest probability ($p_{low}$) in the top-5 vs. log of the probability mass remaining ($p_{rem}$) excluding the top-5.

If the top-5 log probabilities in descending order are $l_{x_i}$ $i \in \{0, 1, 2, 3, 4\}$. Let $x_k$ represent the token (either 'yes' or 'no') that is present in the top-5, with its corresponding log probability denoted as $l_{x_k}$.

$$l_{x_c} = log(min(p_{low}, p_{rem})) \text{ where } p_{low} = exp(l_{x_4}) , \; p_{rem} = 1 - \sum_{i=0}^{4} exp(l_{x_i})$$

$$\text{L3Score} = \text{softmax}(x)_{yes} = \begin{cases} \frac{\exp(l_{x_k})}{\exp(l_{x_k})+\exp(l_{x_c})} & \text{if } x_k = \text{yes} \\ 1 - \text{softmax}(x)_{no} = \frac{\exp(l_{x_c})}{\exp(l_{x_k})+\exp(l_{x_c})} & \text{if } x_k = \text{no} \end{cases} \tag{2}$$

Figure 4 shows QA evaluations where L3Score is more appropriate than ROUGE and BERTScore. We provide additional examples comparing L3Score with other LLM-based metrics in Appendix H.

## 5 Experiments

### 5.1 Experimental Seutp

We evaluate six state-of-the-art closed-source and six open-source models on the test sets of SPIQA for three different task setups: direct QA with figures and tables, direct QA with full paper, and CoT QA. For the long-context models like Gemini [72, 63], GPT [1, 54] and Claude 3 [2], we input all

---
[3]We choose $n = 5$ for top-$n$ log probabilities as this is the highest value of $n$ returned by the OpenAI API.

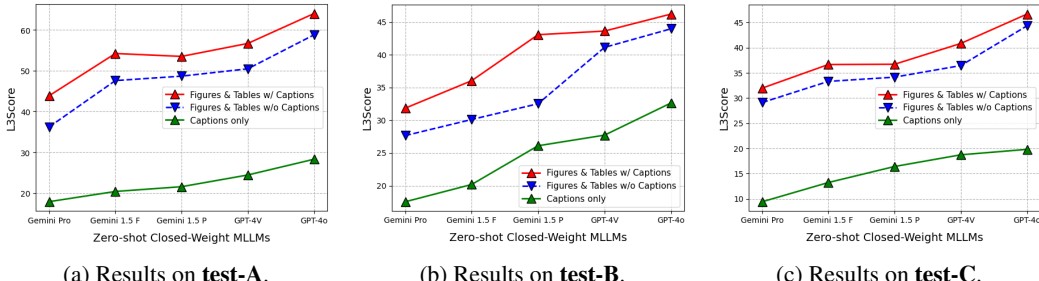

| (a) Results on **test-A**. | (b) Results on **test-B**. | (c) Results on **test-C**. |

Figure 3: **Ablation on the importance of captions in the QA task.** All Gemini and GPT variants suffer when captions are omitted. All numbers are for direct QA with figures and tables.

the images together and ask the model to answer questions. Since most open-weight models can only take one image in a single query, we ask the question and show every image one by one in a multi-turn setup, and then the model answers. We use full-resolution images for models with high context length and 224px images for low context length. For evaluating the free-form answers, we report five different metrics for comprehensive analysis - METEOR [4], CIDEr [77], ROUGE-L [39], BERTScore F1 [88] and the proposed L3Score. For the CoT QA task, we also report the top-1 accuracy for retrieving the helpful images to answer the question. We further elaborate on the importance of comprehending the captions with the figures and tables and report granular results on different figure subcategories.

**Fine-tuning Details.** We fine-tune two open-sourced models, InstructBLIP [13] and LLaVA 1.5 [40] on SPIQA training set with simple QA prompts, where every training sample contains one reference image, corresponding question and answer. We initialize both models from the publicly available checkpoints[4] and train them for two epochs with LoRA [19] of rank 32. We use AdamW [44] optimizer, a cosine scheduler [43] with a linear warmup for the first 3% steps, a peak learning rate of 1e-5, and a batch size 32. Each training job takes about 4 hours on 8 A6000 GPUs. We do not use the main text from papers during training, as InstructBLIP and LLaVA have a low context length of 2048. We resize the images to 224px for training. The trained models are evaluated for direct QA with figures and tables.

## 5.2 Main Results

We highlight the highest scores among closed and open models in every table with red and blue, respectively, and indicate the performance improvements by fine-tuned models with $\Delta$.

**Direct QA w/ Figures and Tables.** Table 3 reports the performance of all open, closed, and fine-tuned models for direct QA with figures and tables. We also use the captions with the images in this setup. Among the open-sourced systems, InternLM-XC and CogVLM perform slightly better than others. InternLM-XC achieves 18.28 and 12.47 L3Score on test-A and test-B, which are ∼5 points superior to InstructBLIP, SPHINX, LLaVA, and XGen-MM. CogVLM performs better than InternLM-XC on test-C, yielding a L3Score of 12.52. Since these models are solely trained on natural images, in most cases, they are unable to understand complex figures and tables and generate random answers. Among the closed models, GPT-4o performs consistently well on test-A and test-C, producing state-of-the-art results on all metrics. Claude-3 also demonstrates strong performance and achieves the highest L3Score score of 49.54 on test-B, which is 3.32 points higher than GPT-4o. We also observe the variability of traditional QA metrics across different models. For example, on test-C, Gemini 1.5 Pro achieves 1.0 point higher METEOR score than Gemini Pro Vision but also attains 2.6 and 7.6 points lower ROUGE and CIDEr scores. In contrast, our proposed L3Score metric considers the semantic meaning of answers instead of mere token matching and persistently generates higher scores for better answers.

The fine-tuned InstructBLIP and LLaVA 1.5 obtain a massive improvement of 28 and 26 point L3Score on average over three test sets compared to corresponding zero-shot models. These fine-tuned models perform almost equally well as Gemini Pro Vision, a powerful long-context closed system. Such results signify the effectiveness of our proposed large-scale training corpus, which will pave the way for developing dedicated QA systems for scientific papers in the future.

---

[4]InstructBLIP and LLaVA 1.5

| Method | Tables | | | Figures | | | | | | | | | | | | Overall | | |
|---|---|---|---|---|---|---|---|---|---|---|---|---|---|---|---|---|---|---|
| | | | | Schematics | | | Plots & charts | | | Visualizations | | | Other figures | | | | | |
| | C | B-F1 | L3S | C | B-F1 | L3S | C | B-F1 | L3S | C | B-F1 | L3S | C | B-F1 | L3S | C | B-F1 | L3S |
| *Zero-shot Closed-Weight MLLMs* | | | | | | | | | | | | | | | | | | |
| Gemini Pro Vision [72] | 130.7 | 65.12 | 39.95 | 111.2 | 66.79 | 45.28 | 127.2 | 63.47 | 45.63 | 119.1 | 65.15 | 51.74 | 113.4 | 65.79 | 47.95 | 124.6 | 64.87 | 43.85 |
| Gemini 1.5 Flash [63] | 128.1 | 66.79 | 47.71 | 86.9 | 67.50 | 60.72 | 101.0 | 62.81 | 54.02 | 112.1 | 66.91 | 63.47 | 96.7 | 67.28 | 62.16 | 110.9 | 65.84 | 54.20 |
| Gemini 1.5 Pro [63] | 88.9 | 65.77 | 55.53 | 60.8 | 64.67 | 55.80 | 101.6 | 62.48 | 48.34 | 80.4 | 63.63 | 57.83 | 70.4 | 64.22 | 56.28 | 87.1 | 64.36 | 53.49 |
| GPT-4 Vision [1] | 132.6 | 64.64 | 56.68 | 44.3 | 63.99 | 60.14 | 133.3 | 63.19 | 53.74 | 93.0 | 62.52 | 58.55 | 69.2 | 63.92 | 60.93 | 113.8 | 64.01 | 56.67 |
| GPT-4o [54] | 130.7 | 65.90 | 65.18 | 114.9 | 68.20 | 71.38 | 153.6 | 65.35 | 56.73 | 96.6 | 64.81 | 56.07 | 117.9 | 67.20 | 67.57 | 133.7 | 66.14 | 64.00 |
| *Zero-shot Open-Weight MLLMs* | | | | | | | | | | | | | | | | | | |
| InstructBLIP-7B [13] | 30.6 | 43.59 | 3.68 | 50.9 | 54.48 | 8.28 | 92.2 | 47.11 | 8.92 | 131.1 | 55.63 | 14.81 | 76.7 | 53.44 | 10.26 | 62.6 | 47.70 | 7.50 |
| LLaVA-1.5-7B [40] | 104.9 | 58.00 | 8.90 | 86.4 | 67.59 | 18.09 | 159.2 | 61.02 | 15.68 | 123.5 | 65.87 | 18.03 | 95.4 | 66.53 | 18.11 | 117.8 | 61.61 | 13.86 |
| XGen-MM [64] | 79.0 | 55.74 | 7.42 | 99.5 | 58.99 | 17.35 | 200.6 | 60.06 | 17.36 | 157.7 | 64.61 | 19.91 | 122.3 | 60.37 | 18.70 | 127.0 | 58.41 | 13.74 |
| InternLM-XC [15] | 93.1 | 54.40 | 13.09 | 66.4 | 60.49 | 25.79 | 38.5 | 44.25 | 17.48 | 98.3 | 61.5 | 24.98 | 78.9 | 60.66 | 25.75 | 73.7 | 53.57 | 18.28 |
| CogVLM [80] | 79.5 | 52.10 | 13.89 | 50.9 | 51.80 | 19.47 | 47.3 | 50.25 | 18.22 | 32.3 | 49.84 | 20.89 | 43.4 | 51.03 | 20.30 | 59.2 | 51.24 | 16.89 |

Table 6: **Performace on tables and figure subcategories.** All numbers are for direct QA with figures and tables on test-A. C: CIDEr, B-F1: BERTScore F1 and L3S: L3Score.

**Direct QA w/ Full Paper.** In this task, we provide the full text of the paper, including figures, tables, and captions, and ask the models to answer a question directly. While the complete text aids in comprehending the article, it also necessitates long-context reasoning capabilities, as scientific papers are typically 8-10 pages long. As shown in Table 5, powerful multimodal models such as Gemini 1.5, GPT-4o, and GPT-4 Vision exhibit significant performance improvements when using the full paper. For example, Gemini 1.5 Pro achieves L3Score improvements of 8.31, 2.24, and 2.76 points with the full text compared to using only figures and tables. This demonstrates the potential of leveraging the full text, which we make available with SPIQA, to develop more advanced models in the future.

**CoT QA.** Table 5 presents the performance of the Gemini and GPT models on the CoT QA task. In this task, the system first retrieves reference images and then generates the answer. The CoT prompt guides the model through step-by-step reasoning, which often results in better responses. For instance, GPT-4 Vision shows an increase of 6.70, 1.73, and 2.98 in L3Score when using CoT prompts compared to direct QA. Similar improvements are observed in most cases for the Gemini models. For simplicity, we perform CoT QA using only figures and tables.

## 5.3 Ablation studies and performance on sub-categories

**Captions are helpful.** In scientific papers, figures and tables are often accompanied by detailed and informative captions. Figure 3 shows that all Gemini and GPT variants experience significant performance drops when captions are excluded. For instance, the best-performing GPT-4o model suffers a 2-point L3Score decrease on test-A when captions are omitted, which underscores the importance of captions in providing context and enhancing comprehension in QA tasks.

**Figure Subcategories.** Table 6 shows the performance of baseline models on tables and figures in test-A. Baseline models perform well on schematic diagrams but struggle with plots, charts, and visualizations. For example, GPT-4o scores 71.38 on schematics but only 56.73 and 56.07 on plots & charts, and visualizations, respectively. Such results indicate the need for improved systems to comprehend scientific diagrams requiring mathematical reasoning.

**Tables harder for open models.** Table 6 indicates open-source models struggle with tables, scoring lower than their overall performance. All closed-source models perform well. Notably, GPT-4o achieves a 65.18 L3Score on tables, outperforming GPT-4 Vision, the second best by 8.50 points.

**Image resolution matters.** Appendix Figure B.1 includes performance of the closed models on images of different resolutions. Full resolution images result in consistently better performance.

## 5.4 Human Evaluation

We conduct a human evaluation on a 20% subset of test-A to establish a human performance baseline for the CoT QA task in Table 7. The diversity of research domains in the SPIQA dataset poses a challenge in obtaining a reliable human upper bound, as it requires broad expertise to answer

| Method | CoT QA on SPIQA test-A | | | | | |
|---|---|---|---|---|---|---|
| | Ret. Acc. | M | R-L | C | B-F1 | L3S |
| Human Eval | 94.89 | 28.2 | 44.4 | 130.9 | 69.90 | 68.10 |
| Gemini 1.5 Flash [63] | 87.14 | 26.0 | 39.1 | 123.2 | 67.10 | 58.72 |
| Gemini 1.5 Pro [63] | 86.58 | 26.2 | 39.0 | 103.4 | 66.75 | 65.16 |
| GPT-4 Vision [1] | 84.30 | 26.0 | 38.6 | 110.1 | 66.94 | 64.55 |
| GPT-4o [54] | 86.58 | 27.4 | 43.7 | 130.2 | 68.93 | 66.36 |

Table 7: **Human evaluation on a 20% subset of test-A on CoT QA.** We also report the scores of baseline models on the same subset of test-A to contextualize their performance compared to human capability.

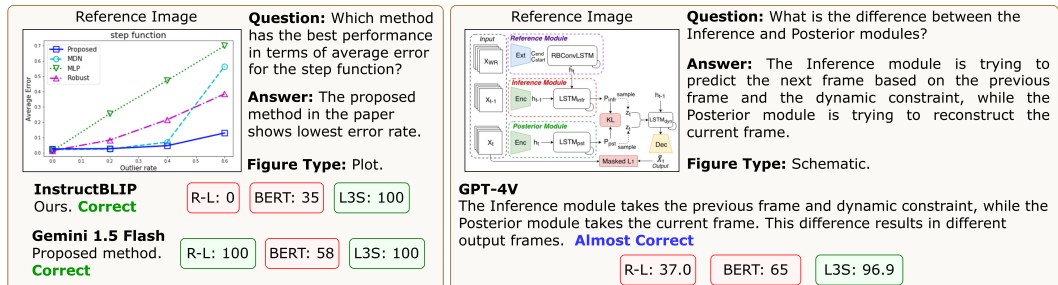

Figure 4: **Example questions, ground-truth answers, and responses by different baseline models.** Both QAs belong to testA. Metrics colored in green denote correct evaluations, while those in red indicate incorrect scoring. R-L: ROUGE-L, BERT: BERTScore, L3S: L3Score.

questions accurately. Additionally, the free-form response format yields varied answers depending on the evaluator's knowledge and understanding of the figures and tables. To study these, we employ an experienced AI/ML researcher to evaluate a fraction of test-A, using the same setup as the large multimodal models. The human evaluator identifies relevant figures and tables and provides a free-form answer to the question, achieving a top-1 accuracy of 94.89% in retrieving the correct figure and outperforming the best model by approximately 8%. Unsurprisingly, on other automated metrics, the performance of human and LLMs are similar due to a single ground truth answer. To establish a more robust human baseline, future work should involve multiple human evaluators with diverse expertise.

## 5.5 Qualitative Results and Additional Analysis

Figure 4 illustrates two example questions, ground-truth answers, and outputs from various baseline models. For the plot-based question, both InstructBLIP and Gemini 1.5 Flash produce correct responses. However, only our proposed L3Score evaluates them correctly in both cases. Notably, InstructBLIP's response ('*Ours*') correctly answers the question despite not matching the ground-truth. Traditional metrics like ROUGE-L and BERTScore fail to evaluate such responses, scoring them as 0. For the schematic-based question, the ground-truth is long and descriptive. GPT-4 Vision generates a significantly correct response, and L3Score accurately evaluates it with a score of 96.9. We observe that the best-performing GPT-4o struggles with complex plots, charts, and tables. More such error cases are detailed in Appendix H.

**Analysis of L3Score.** Table 8 shows that the proposed L3Score metric correlates well with existing automated metrics. Appendix B presents additional analysis of the proposed metric. E.g., we compute L3Score using different LLMs (GPT 3.5, 4o, Gemini) and find it to be consistent when comparing the outputs of different baselines although the absolute score itself may be different depending on the model used for computing the L3Score.

| Spearman's $\rho$ | M | R-L | C | B-F1 |
|---|---|---|---|---|
| w. L3Score | 0.71 | 0.72 | 0.69 | 0.78 |

Table 8: **Correlation of L3Score with existing metrics** for free-form QA evaluation.

## 6 Conclusion

We introduce SPIQA (**S**cientific **P**aper **I**mage **Q**uestion **A**nswering), the first large-scale QA dataset specifically designed to interpret complex figures and tables within the context of scientific papers. Additionally, we propose LLMLogScore(L3Score), an improved metric for free-form QA that accurately analyzes the semantic context of candidate answers relative to the ground truth. Through extensive experiments with 12 prominent foundational models, we evaluate their ability to comprehend the nuanced aspects of research articles. Furthermore, fine-tuning two open-source systems, LLaVA and InstructBLIP, on the SPIQA training set results in significant improvements over zero-shot evaluations, indicating promising avenues for designing specialized systems for scientific QA in the future. Our work lays the foundation for developing advanced QA systems that effectively understand and analyze scientific documents, driving further research in this critical area.

**Limitations and Societal Impact.** We acknowledge that SPIQA is designed for research purposes and should not be regarded as a comprehensive dataset, as it is restricted to computer science papers. Models trained on our dataset may exhibit biases towards specific topics within this domain and may not perform well on other scientific literature. Extending SPIQA to encompass other scientific domains remains a future prospect.

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

# A    Dataset and Code Release

Following NeurIPS Dataset and Benchmark Track guidelines, we publicly release the SPIQA dataset on Hugging Face: https://huggingface.co/datasets/google/spiqa. Our evaluation and metric computation scripts, along with responses from all closed- and open-source models, are accessible in the GitHub repository: https://github.com/google/spiqa. SPIQA is available under the Creative Commons Attribution License (CC BY 4.0). The evaluation code and library for L3Score are available in the Github repository under Apache 2.0 license. We include the dataset card and README for the resources.

# B    Additional studies on L3Score and ablations.

| Method | L3Score on SPIQA test-B | | |
| --- | --- | --- | --- |
| | Gemini Pro [72] | GPT-3.5 T [53] | GPT-4o [54] |
| Gemini Pro Vision [72] | 47.53 | 28.14 | 33.10 |
| Gemini 1.5 Flash [63] | 54.55 | 32.76 | 36.04 |
| Gemini 1.5 Pro [63] | 56.78 | 35.15 | 43.27 |
| GPT-4 Vision [1] | 65.83 | 38.30 | 43.62 |
| GPT-4o [54] | 68.61 | 39.58 | 46.22 |
| InstructBLIP-7B [13] | 18.82 | 7.26 | 7.07 |
| LLaVA-1.5-7B [40] | 20.69 | 10.71 | 9.63 |
| XGen-MM [64] | 15.55 | 7.46 | 8.18 |
| InternLM-XC [15] | 20.51 | 14.09 | 12.47 |
| CogVLM [80] | 18.44 | 10.01 | 9.60 |

Table B.1: **Computation of L3Score using different LLMs.** While the absolute values of L3Score vary depending on the choice of LLM, the relative changes in between different models remain consistent. All numbers are for direct QA with figures and tables on test-B.

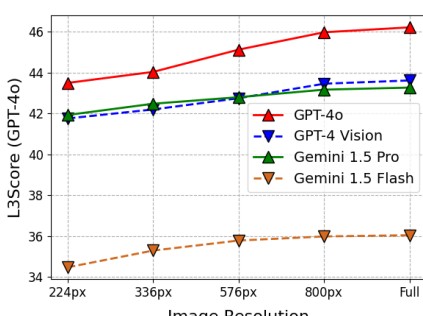

Figure B.1: **Ablation on the input image resolution.** L3Score improves with increasing resolution of images (figures and tables). All numbers are for direct QA with figures and tables on test-B.

**L3Score with Different LLMs.** Table B.1 compares the L3Score across various LLMs. Fig. F.1 shows the prompt we used to compute the L3Score to evaluate responses from models. We observe that while the absolute L3Score value varies depending on the chosen LLM, the relative changes in the scores remain consistent. For instance, when the L3Score is computed using GPT-3.5 Turbo, GPT-4o scores 11.44 points higher than Gemini Pro Vision. Using GPT-4o to compute the L3Score, the difference increases to 13.12 points. However, GPT-4o achieves approximately 40% better scores in both cases than Gemini Pro Vision. Different LLMs have varying vocabularies, which can cause slight differences in their log-likelihood probabilities. Notably, L3Score can be computed with any LLM that provides log probabilities for different output tokens. We recommend users consistently use the same LLM across all evaluations to ensure proper score comparison. Except for Table B.1, we always use GPT-4o when calculating L3Score, which is currently one of the most powerful and reasonably affordable publicly available LLMs.

**Image Resolution.** Figure B.1 demonstrates the performance improvements of different Gemini and GPT-4 systems when using higher image resolutions. Generally, higher resolution images lead to increased L3Scores. However, larger input images result in more input tokens, making the LLM call more expensive. In our main experiments, we always use full-resolution images for the closed models and 224px images for the open models.

| Method | Mode of Evaluation | Direct QA on SPIQA test-A | | | | |
| --- | --- | --- | --- | --- | --- | --- |
| | | M | R-L | C | B-F1 | L3S |
| | Zero-shot | 22.6 | 34.7 | 117.8 | 61.61 | 13.86 |
| LLaVA 1.5 | Unsupervised | 22.6 | 35.1 | 118.2 | 61.53 | 14.43 |
| 7B [40] | Supervised | 23.8 | 36.0 | 121.2 | 63.74 | 45.45 |
| | Unsupervised + Supervised | 24.0 | 36.4 | 121.7 | 63.92 | 46.11 |

Table B.2: **Ablation on fine-tuning strategy.** Supervised fine-tuning yields significantly more improvements, showing the usefulness of the model-generated QAs on SPIQA training set.

**Supervised vs. Unsupervised Tuning on SPIQA Training Set.** To further understand the quality of model-generated QAs, we conduct an ablation study to investigate the impact of unsupervised training using figure-caption pairs from the SPIQA training set and report the results in Table B.2. During unsupervised training, we ask the model to describe a given figure, using the caption as ground truth. This approach does not require the generated QAs. As shown in the first two rows of Table B.2, such unsupervised training yields only marginal improvement over zero-shot evaluation on test-A.

In contrast, supervised fine-tuning using QAs significantly enhances performance, demonstrating the value of generated annotations for advanced question-answering. Furthermore, we found that combining unsupervised figure-caption pre-training followed by QA-based fine-tuning leads to additional improvements, highlighting the potential benefits of large-scale QA data for developing specialized QA systems for scientific papers in the future.

## C SPIQA vs. Existing Scientific QA Datasets

Manually generating free-form, open-ended questions and answers on scientific articles is a demanding process in terms of cost and time, requiring expert annotators with detailed domain-specific knowledge. Many existing scientific QA benchmarks typically follow cloze-style question generation to bypass such costly annotation procedures. This approach removes a named entity from a single sentence, and the task is to guess the missing entity after reading the preceding passage. For example, BioRead [58] uses the full text of unlabeled biomedical articles from PubMed[5] and utilizes Metamap [3] to annotate biomedical entities and generate cloze-style questions. BioRead extracts sequences of 21 sentences from the articles, using the first 20 sentences as a passage and the last sentence as a question. BioMRC [59] improves and cleans the BioRead corpus by avoiding cross-section extraction and excluding text from references, captions, and tables. BioMRC uses 25 million abstracts from Pubtator[6] [81] and DNORM's [33] biomedical entity annotations to generate 812K cloze-form questions. In a similar line, emrQA [56] develops the first patient-specific electronic medical records (EMR) QA dataset to help physicians quickly find information from clinical notes collected from the i2b2 dataset[7]. MedHop [82] constructs a corpus for detecting Drug-Drug Interactions (DDIs) from multiple source documents using 24 million paper abstracts collected from the Medline 2016 release[8]. Similarly, CliCR [69] creates 105K gap-filling queries based on 12K clinical case reports collected from BMJ Case Reports[9]. However, these cloze-form datasets fail to reflect real-world scenarios where the readers often have open-ended questions when reading a long scientific research paper.

BioAsq [3, 30] is one of the first scientific QA benchmarks that employed human experts to annotate free-form questions based on abstracts of biomedical articles collected from the PubMed corpus. Since BioAsq primarily contains simple factual questions, PubMedQA [21] provides a more comprehensive biomedical QA benchmark that requires detailed reasoning over article abstracts. QASPER [14] introduces the first QA dataset outside the biomedical field by collecting 1,585 natural language processing (NLP) papers from the S2ORC corpus[10] and generating 5,049 open-ended questions. Graduate students and freelancers with NLP expertise were recruited as annotators and were provided with the titles and abstracts of the research papers. They were instructed to write questions that could not be answered from the title and abstract but were expected to be addressed somewhere in the paper. Subsequently, experts reviewed the annotated queries and the entire paper to provide the answers. Due to this two-step annotation procedure, many questions in QASPER remain unanswerable from the paper. Additionally, most answerable questions in QASPER can be responded to with a binary yes/no answer or a short extractive span, making the questions relatively easy and not requiring a deep understanding of the entire scientific paper. ArgSciChat [66] proposes the first dataset with argumentative dialogues based on NLP papers, consisting of multi-turn conversations between scientists. However, ArgSciChat contains only 41 dialogues over 20 scientific papers.

Closest to our work, QASA [35] contains 1,798 open-ended and detailed questions on 112 AI/ML papers, where annotators read the full paper text when writing the questions. The answers to these questions are multi-faceted and long-form, written by AI/ML experts or the actual authors of the corresponding papers. Due to the expensive annotation scheme, despite maintaining high quality, the QASA benchmark is small-scale and the questions do not require understanding complex figures and diagrams. Our proposed SPIQA dataset differs from existing scientific QA benchmarks in three key aspects: ($i$) We introduce the first large-scale, free-form, open-ended scientific QA dataset covering all domains of computer science. ($ii$) The questions and answers in SPIQA require a holistic understanding of complex figures and tables, along with the full textual content of the papers. ($iii$) In addition to the direct QA setup, we propose a novel Chain-of-Thought (CoT) QA paradigm, where

---

[5]PubMed: https://www.ncbi.nlm.nih.gov/pmc/

[6]Pubtator: https://www.ncbi.nlm.nih.gov/research/pubtator3/

[7]i2b2 datasets: https://www.i2b2.org/NLP/DataSets/

[8]Medline: https://www.nlm.nih.gov/medline/medline_home.html

[9]BMJ Case Report: http://casereports.bmj.com/

[10]S2ORC: https://github.com/allenai/s2orc

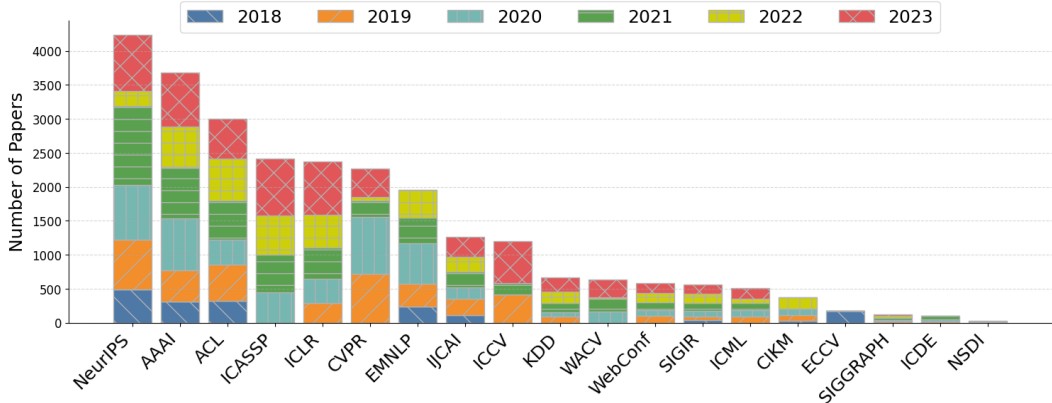

Figure D.1: **Source of Collected Papers.** SPIQA comprises a total of 25,859 papers published in 19 different top-tier conferences between 2018 and 2023, covering various domains of computer science.

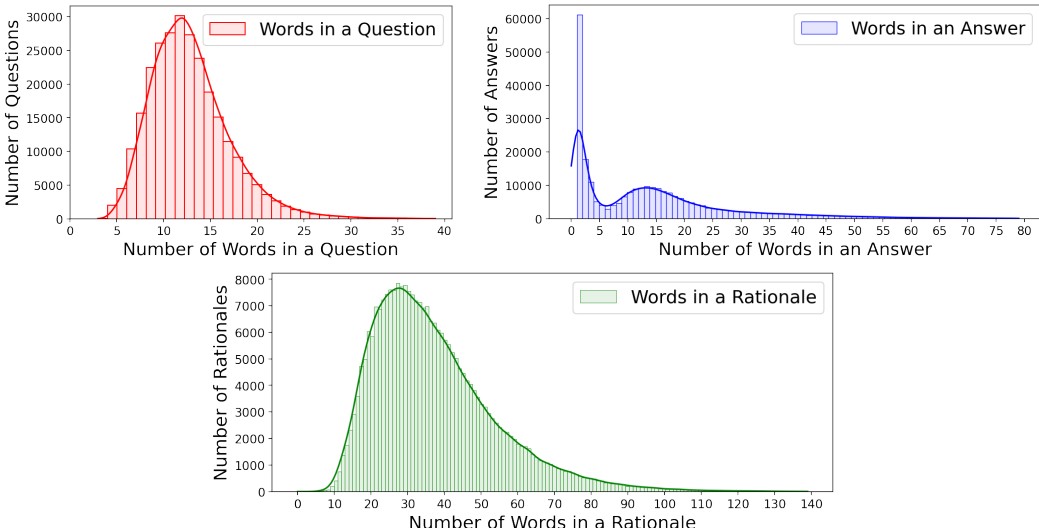

Figure D.2: **Distribution of number of words in questions, answers and rationales in SPIQA.** We observe a large variety in the length of QAs, which indicates a diverse range of patterns in SPIQA.

models first identify helpful figures and tables, and then generate the answer. This step-by-step QA pipeline helps evaluate the fine-grained reasoning capabilities of the baseline systems. Table 1 provides an extensive comparison of SPIQA with all available scientific QA benchmarks.

# D    Additional Dataset Analysis

As described in Section 3.2, SPIQA consists of 25,859 papers published in 19 different top-tier conferences between 2018 and 2023, covering various domains of computer science. Figure 2 categorizes the source conferences into four broad groups: ($i$) AI/ML: This category contributes 46% of SPIQA, with conferences such as NeurIPS, ICLR, ICML, AAAI, and IJCAI. ($ii$) Natural language processing (NLP): Conferences like ACL and EMNLP make up 19% of the dataset. ($iii$) Computer vision and computer graphics: This category includes CVPR, ICCV, ECCV, WACV, and SIGGRAPH, contributing 17% of the papers. ($iv$) Other computer science domains: These include information retrieval (SIGIR, CIKM), databases (ICDE), networking (WebConf, NSDI), data mining (KDD), and audio and signal processing (ICASSP), collectively covering the remaining 18% of SPIQA. Figure D.1 illustrates the number of papers from each conference and each year between 2018 and 2023.

After collecting the papers, we generated 270,194 question-answer-rationale triplets using Gemini 1.5 Pro, focusing on the figures, tables, and text of the scientific articles. The average lengths of the questions, answers, and rationales are 12.98, 14.56, and 37.42 words, respectively. We also observed high variances: 20.47 for questions, 243.29 for answers, and 468.91 for rationales. As shown in Figure D.2, approximately 36.62% of answers contain 5 words or fewer, 56.70% contain between 6

| Method | SPIQA test-A | | | | SPIQA test-B | | | | SPIQA test-C | | | |
|---|---|---|---|---|---|---|---|---|---|---|---|---|
| | B@1 | B@2 | B@3 | B@4 | B@1 | B@2 | B@3 | B@4 | B@1 | B@2 | B@3 | B@4 |
| *Zero-shot Closed-Weight MLLMs* | | | | | | | | | | | | |
| Gemini Pro Vision [72] | 36.3 | 27.8 | 22.7 | 18.8 | 23.1 | 12.2 | 7.3 | 4.7 | 15.7 | 9.0 | 6.1 | 4.5 |
| Gemini 1.5 Flash [63] | 35.4 | 27.3 | 22.2 | 18.4 | 26.8 | 13.9 | 8.1 | 5.1 | 16.9 | 10.8 | 8.0 | 6.4 |
| Gemini 1.5 Pro [63] | 35.9 | 26.7 | 21.1 | 17.0 | 24.2 | 12.6 | 7.4 | 4.6 | 15.5 | 9.2 | 6.4 | 4.9 |
| Claude 3 (Opus) [2] | 37.2 | 30.2 | 23.7 | 18.7 | 21.9 | 11.6 | 6.9 | 4.4 | 16.8 | 10.6 | 7.9 | 6.5 |
| GPT-4 Vision [1] | 28.7 | 21.3 | 16.9 | 13.8 | 22.0 | 11.3 | 6.7 | 4.5 | 15.1 | 9.4 | 6.9 | 5.5 |
| GPT-4o [54] | 40.5 | 31.4 | 25.5 | 21.2 | 23.3 | 12.6 | 7.9 | 5.3 | 20.8 | 13.1 | 9.6 | 7.6 |
| *Zero-shot Open-Weight MLLMs* | | | | | | | | | | | | |
| InstructBLIP-7B [13] | 15.2 | 11.9 | 10.0 | 8.5 | 1.6 | 0.9 | 0.6 | 0.5 | 5.4 | 2.5 | 1.5 | 0.9 |
| LLaVA-1.5-7B [40] | 35.1 | 27.6 | 23.1 | 19.7 | 16.9 | 8.8 | 5.3 | 3.6 | 12.8 | 6.0 | 3.4 | 2.1 |
| XGen-MM [64] | 31.1 | 24.9 | 21.1 | 18.2 | 2.8 | 1.4 | 0.9 | 0.6 | 8.0 | 4.6 | 3.3 | 2.5 |
| InternLM-XC [15] | 35.4 | 27.5 | 22.6 | 19.1 | 15.6 | 8.3 | 5.3 | 3.7 | 16.2 | 9.1 | 6.2 | 4.6 |
| CogVLM [80] | 33.9 | 26.5 | 21.8 | 18.4 | 15.8 | 8.4 | 5.3 | 3.8 | 15.8 | 8.7 | 5.8 | 4.2 |
| *Fine-tuned MLLMs* | | | | | | | | | | | | |
| InstructBLIP-7B [13] | 34.1 | 26.0 | 22.3 | 18.5 | 15.7 | 9.5 | 5.8 | 4.2 | 13.1 | 8.3 | 6.3 | 4.4 |
| $\Delta_{\text{InstructBLIP-7B FT - ZS}}$ | 18.9↑ | 14.1↑ | 12.3↑ | 10.0↑ | 14.1↑ | 8.6↑ | 5.2↑ | 3.7↑ | 7.7↑ | 5.8↑ | 4.8↑ | 3.5↑ |
| LLaVA-1.5-7B [40] | 38.0 | 29.6 | 24.7 | 21.0 | 22.6 | 11.2 | 7.5 | 5.1 | 16.9 | 10.1 | 7.0 | 4.6 |
| $\Delta_{\text{LLaVA-1.5-7B FT - ZS}}$ | 2.9↑ | 2.0↑ | 1.6↑ | 1.3↑ | 5.7↑ | 2.4↑ | 2.2↑ | 1.5↑ | 4.1↑ | 4.1↑ | 3.6↑ | 2.5↑ |

Table E.1: **Performance of zero-shot and fine-tuned systems on direct QA with figures and tables in terms of BLEU scores.** B@1: BLEU@1, B@2: BLEU@2, B@3: BLEU@3, B@4:BLEU@4. We highlight the highest scores among closed and open models in every table with red and blue, respectively. $\Delta$ shows improvements after fine-tuning.

| Method | SPIQA test-A | | | | | SPIQA test-B | | | | | SPIQA test-C | | | | |
|---|---|---|---|---|---|---|---|---|---|---|---|---|---|---|---|
| | Ret. Acc. | QA B@1 | B@2 | B@3 | B@4 | Ret. Acc. | QA B@1 | B@2 | B@3 | B@4 | Ret. Acc. | QA B@1 | B@2 | B@3 | B@4 |
| Gemini 1.5 Flash [63] | – | 35.4 | 27.3 | 22.2 | 18.4 | – | 26.8 | 13.9 | 8.1 | 5.1 | – | 16.9 | 10.8 | 8.0 | 6.4 |
| w/ Full Paper | – | 37.5 | 29.0 | 23.7 | 19.9 | – | 30.3 | 17.7 | 11.9 | 8.6 | – | 17.5 | 11.2 | 8.7 | 6.7 |
| w/ CoT | 86.18 | 37.1 | 28.5 | 23.2 | 19.2 | 57.45 | 26.0 | 13.5 | 7.8 | 4.8 | 69.37 | 15.4 | 9.6 | 7.0 | 5.5 |
| Gemini 1.5 Pro [63] | – | 35.9 | 26.7 | 21.1 | 17.0 | – | 24.2 | 12.6 | 7.4 | 4.6 | – | 15.5 | 9.2 | 6.4 | 4.9 |
| w/ Full Paper | – | 38.0 | 28.2 | 23.0 | 18.3 | – | 27.5 | 15.3 | 9.6 | 6.5 | – | 16.3 | 9.7 | 7.3 | 5.5 |
| w/ CoT | 85.88 | 37.1 | 27.8 | 22.1 | 17.9 | 62.28 | 24.3 | 12.3 | 7.1 | 4.4 | 70.79 | 16.8 | 10.9 | 8.0 | 6.4 |
| GPT-4 Vision [1] | – | 28.7 | 21.3 | 16.9 | 13.8 | – | 22.0 | 11.3 | 6.7 | 4.5 | – | 15.1 | 9.4 | 6.9 | 5.5 |
| w/ Full Paper | – | 33.1 | 25.4 | 19.0 | 16.3 | – | 25.3 | 14.4 | 9.3 | 6.5 | – | 15.6 | 9.8 | 7.5 | 6.0 |
| w/ CoT | 83.25 | 34.3 | 26.0 | 20.9 | 17.2 | 60.45 | 26.8 | 14.0 | 8.5 | 5.7 | 66.73 | 15.0 | 9.2 | 6.7 | 5.2 |
| GPT-4o [54] | – | 40.5 | 31.4 | 25.5 | 21.2 | – | 23.3 | 12.6 | 7.9 | 5.3 | – | 20.8 | 13.1 | 9.6 | 7.6 |
| w/ Full Paper | – | 41.3 | 32.1 | 26.2 | 22.0 | – | 31.7 | 19.4 | 13.8 | 10.7 | – | 21.6 | 14.1 | 10.4 | 7.9 |
| w/ CoT | 85.58 | 40.9 | 31.8 | 26.1 | 21.9 | 63.63 | 24.1 | 13.1 | 8.2 | 5.5 | 70.38 | 18.9 | 11.9 | 8.7 | 6.9 |

Table E.2: **Performance on direct QA with full paper and CoT QA in terms of BLEU scores.** Both tasks help to improve the performance of Gemini and GPT4 models over direct QA with figures and tables. B@1: BLEU@1, B@2: BLEU@2, B@3: BLEU@3, B@4:BLEU@4. We highlight the highest scores in every test split with red.

and 40 words, and the remaining answers contain more than 40 words. This distribution demonstrates the presence of both direct and descriptive or explanatory QAs in SPIQA. The number of words in questions and rationales follows a long-tail normal distribution. Although we do not use the rationales in our experiments, we are releasing them for future research.

# E    Additional Quantitative Results

Tables E.1 and E.2 report the results for three different tasks: direct QA with figures and tables, direct QA with the full paper, and CoT QA, using various BLEU metrics. Similar to Tables 3 and 4 of the main paper, GPT-4o achieves state-of-the-art scores on test-A and test-C. Gemini 1.5 Flash performs particularly well on test-B, achieving a 26.8 BLEU@1 score, which is more than 2 points higher than any other model. Open-source models generally underperform compared to closed-source models, primarily because they are trained on natural images.

After fine-tuning on the training set, both InstructBLIP-7B and LLaVA-1.5-7B show significant performance improvements. InstructBLIP-7B achieves an average BLEU@1 improvement of 13.56

points across the three datasets, while LLaVA-1.5-7B gains an average of 4.22 points BLEU@1 score. Fine-tuning with scientific diagrams enhances these models' ability to comprehend the questions, highlighting the potential importance of our training set for building powerful, specialized systems for scientific QA in the future.

CoT prompts and full-text input enhance the QA performance of various Gemini and GPT-4 systems. On test-A, GPT-4o with the full paper achieves a 41.3 BLEU@1 score, which is 0.8 points higher than its direct QA performance. The GPT-4 Vision models gain an impressive 5.6 points in BLEU@1 score when using CoT compared to direct QA. Similar trends are observed in the other two test splits. The step-by-step fine-grained CoT reasoning and long-context understanding ability of Gemini and GPT-4 models with the full paper contribute to these improved results.

## F  Prompt for L3Score Computation

Fig. F.1 shows the prompt used for computing the proposed LLMLogScore (L3Score) metric based on the log-likelihood of the models responses to binary yes, no questions. We use it to measure similarity of the model predicted answers to a given ground truth answer.

```
You are given a question , ground - truth answer , and a candidate
    answer .

Question : <question >
Ground - truth answer : <GT >
Candidate answer : <answer >

Is the semantic meaning of the ground - truth and candidate answers
    similar ? Answer in one word - Yes or No.
```

Figure F.1: **Prompt Used for Computing L3Score.** We provide the ground-truth and candidate answers and ask the LLM to determine if their semantic meanings are preserved in the context of the question.

## G  Detailed Annotation Guidelines and User Interfaces

The goal of the SPIQA test set is to assist the evaluation of multimodal models on robust understanding of research articles. We prompt the LLM (Gemini 1.5 pro) to generate questions based on a given image. The prompt we use is shown in Figure G.1 and G.2. After generating questions on all papers (≈26k papers, ≈270k images), we subset 200 papers as test set and filter to retain higher quality questions more pertinent to the research article. In the filtering process, we annotate which figures and tables help answer a question to evaluate the grounding and CoT reasoning capabilities of large multimodal systems. The UI used for annotation is shown in Fig. G.3.

We manually verified the quality of the generated question and answer pairs using the following criterion:

1. Answering the question would require understanding of the figure or table and may require domain knowledge.

2. The generated answer is correct and to the point.

3. The question is neither too trivial nor too specific to the figure or table (e.g., avoiding questions like *'What does the blue line in Figure 1 represent?'* or *'How many rows are there in Table 2?'* for being trivial)

4. If two or more questions from a paper are similar, keep one.

5. If the question is entirely based on the passage i.e. cannot be answered from the image, discard the question-answer pair.

6. If the answer is not clear, e.g., the answer says *'It is hard to answer the question based on the given information'* or *'The answer is not evident from the given passage'*, discard the question-answer pair.

7. If the question-answer pair includes phrases like *'Based on the passage,'* modify it because we show all figures and tables to the model at once during evaluation.

```
You are a professor. Generate one question based on the image
and caption to test if a student can interpret and understand the
    image well.
Also classify the figure as "plot", "schematic", "photograph(s)",
    "table" or "others".

Image:
{{ Image }}

Caption: {{ caption }} \

The passage where the figure is referenced is provided below.\

PASSAGE: {{ passage }} \

Construct your questions and corresponding answers. Use this
    format. \
Question: <question that tests understanding of the image.> \
Answer: <Answer to the question based on the passage.> \
Explanation: <How the figure helps answer the question.> \
Figure_type: <"type of figure" where type of figure is one of \
["plot", "schematic", "photograph(s)", "table", "other"]>
```

Figure G.1: **Prompt used for generating questions based on figures in the paper**. We provide the first passage referencing the figure as additional context to the model.

```
You are a professor. Generate one question based on the image
and caption to test if a student can interpret and understand the
    image well.
Also classify the figure as "plot", "schematic", "photograph(s)",
    "table" or "others".

Image:
{{ Image }}

Caption: {{ caption }}

Construct your questions and corresponding answers. Use this
    format.

Question: <question that tests understanding of the image.>
Answer: <Answer to the question based on the passage.>
Explanation: <How the figure helps answer the question.>
Figure_type: <"type of figure" where type of figure is one of
["plot", "schematic", "photograph(s)", "table", "other"]>
```

Figure G.2: **Prompt used for generating questions based on figures in the paper**. We do not include the passage referencing the figure in the prompt when we are unable to extract the figure from the tex source (we use pdffigures to extract the image and do not have the mapping to the corresponding passage).

We initially employed crowd workers at a cost of $22 per hour for filtering the questions. However after a pilot evaluation of 150 questions which were annotated by two different sets of 3 crowd workers and the authors, we found that the crowd workers lacked domain expertise necessary to grasp the nuances in the questions. Example of the pilot UI with the question and response from a crowd worker is shown in Fig. G.4. The filtering of the final SPIQA testA set was done by the authors.

**Paper-ID**

1710.06177v2

**Figure-ID**

1710.06177v2-Figure1-1.png

**Question**

What is the purpose of the Visual Analogy Graph in the VAGER framework?

**Image**

**Caption**

The framework of learning to learn image classifiers. Training Base Classes with VAGER: By training base classes with VAGER, we derive the embeddings of each base class and the transform the embedding into transferred classification parameters by the mapping function learned by VAGER. After training the classifier with new class samples and getting the

**Answer**

The Visual Analogy Graph is used to learn the relationships between different classes of images. It does this by creating a graph where each node represents a class and each edg

**Explanation**

The Visual Analogy Graph is shown in the middle of the figure. It takes the output of the CNNs for each of the training base classes and uses this information to create a graph whei

Consider if the question can be answered from the figure.Should we keep the questions or discard?

○ Keep

○ Discard

Should the question or answer be modified?

○ YES

○ NO

Modified question:

Modified answer:

Figure G.3: **UI used for filtering questions for the SPIQA test-A.** Given the image, question, answer and model's explanation, we ask the annotator if the question can be answered from the figure and whether the question should be kept or discarded. We also ask if the question or answer should be modified and provide text boxes for the annotator to include the modified question and answer.

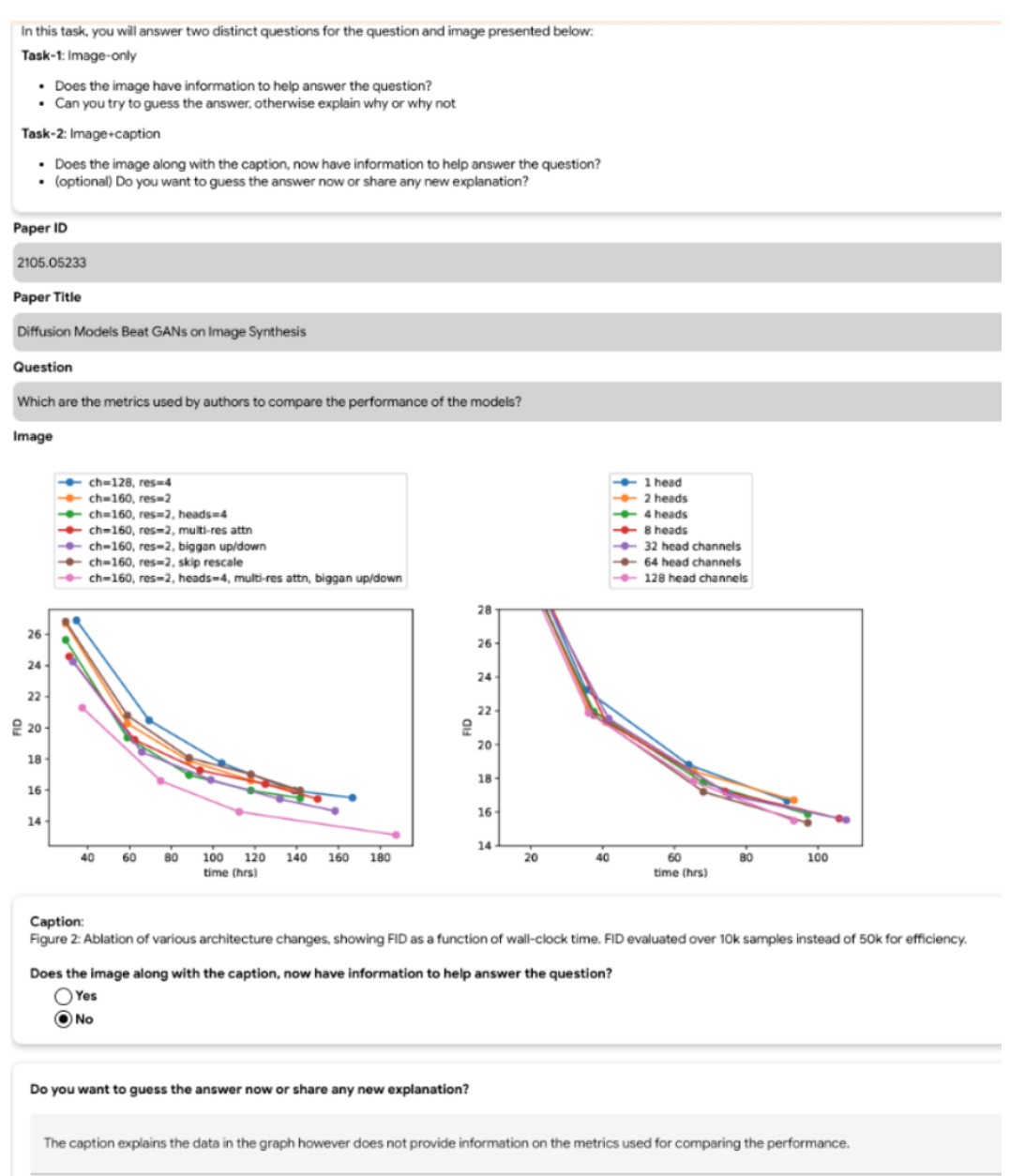

Figure G.4: **Pilot filtering shows that crowd workers lack expertise for the task.** Example question and response within the UI from the pilot set of filtering on SPIQA test-B. Initial trials using crowd workers demonstrated that the task of filtering and identifying pertinent questions also requires expertise in the domain.

# H  Qualitative examples of the task and data in SPIQA

Fig. H.1, H.2 and H.3 show examples of the SPIQA CoT QA task, requiring the analysis of multiple-images when answering questions based on a scientific paper. In the SPIQA dataset, there are on average 10.32 images (figures and tables) per paper. In the CoT QA task, given a question and all the figures and tables, the AI system needs to identify which image is most helpful in answering the question and then provide an answer to the question.

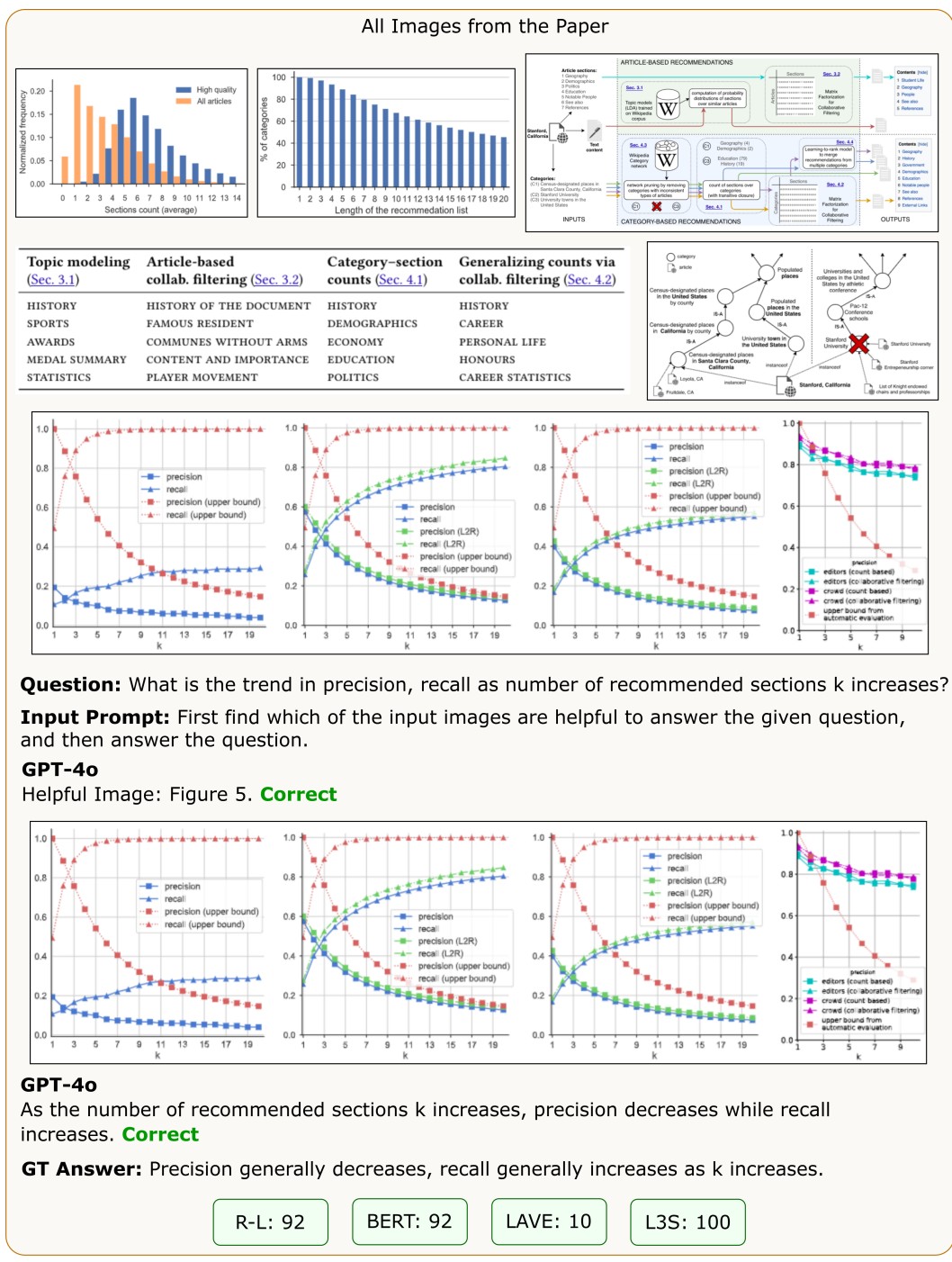

Figure H.1: **Example demonstrating the SPIQA CoT QA task.** Gven a question and all the figures and tables, the AI system needs to identify which image is most helpful in answering the question and then provide an answer to the question. R-L: ROUGE-L, BERT: BERTScore, L3S: L3Score.

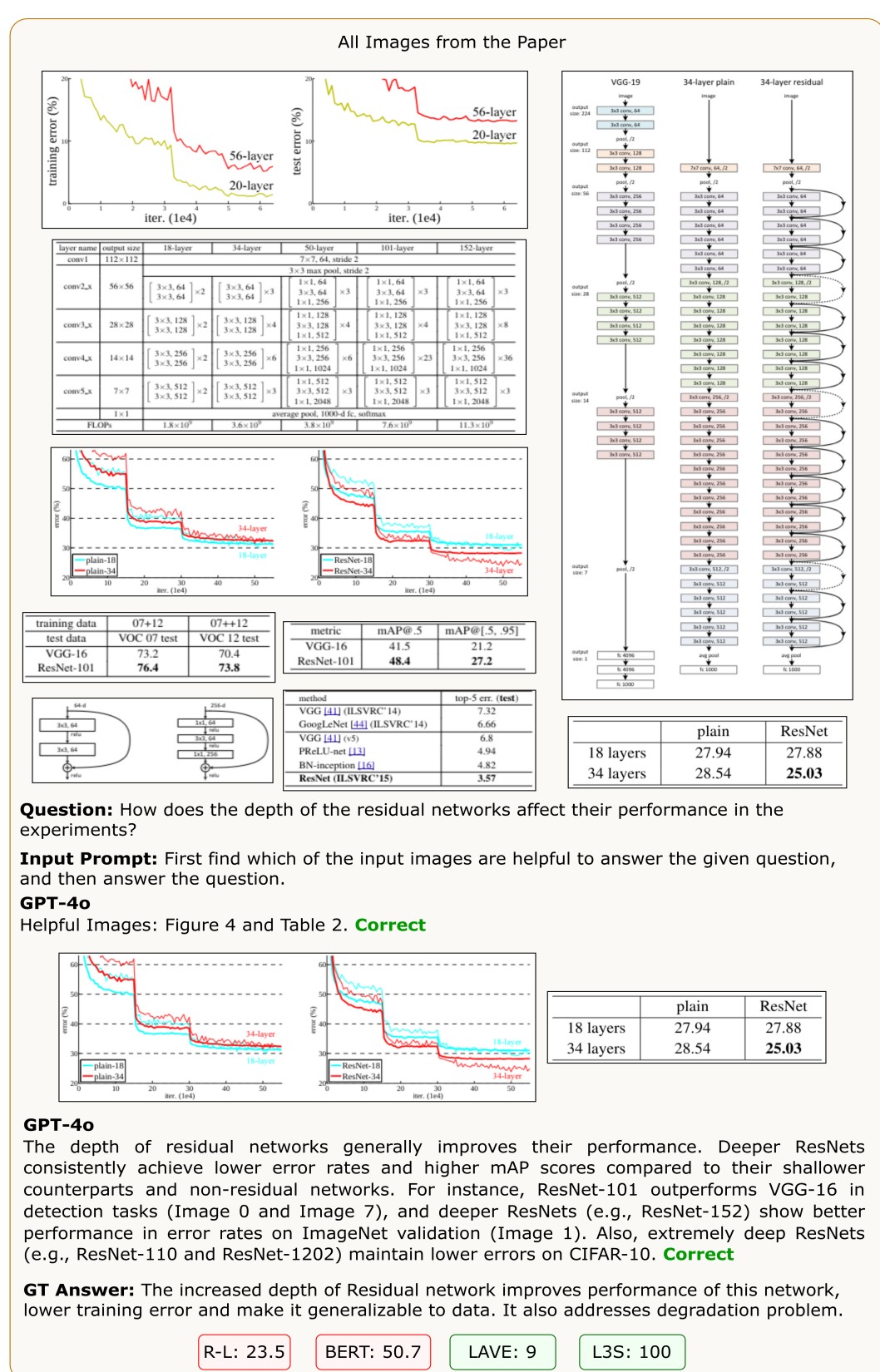

Figure H.2: **Example demonstrating the SPIQA CoT QA task**. Given a question and all the figures and tables, the AI system needs to identify which image is most helpful in answering the question and then provide an answer to the question. R-L: ROUGE-L, BERT: BERTScore, L3S: L3Score.

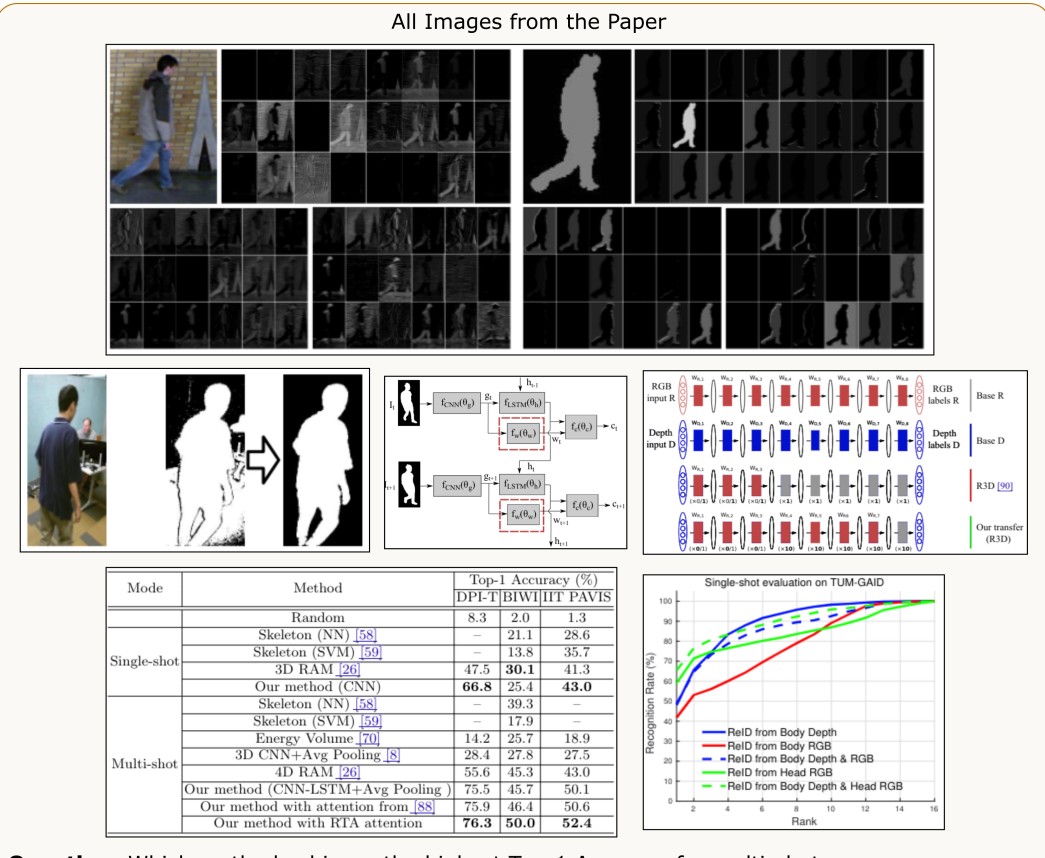

**Question:** Which method achieves the highest Top-1 Accuracy for multi-shot person re-identification on the BIWI dataset, and how does it compare to the best single-shot method?

**Input Prompt:** First find which of the input images are helpful to answer the given question, and then answer the question.

**GPT-4o**
Helpful Image: Table 1. **Correct**

| Mode | Method | Top-1 Accuracy (%) | | |
|---|---|---|---|---|
| | | DPI-T | BIWI | IIT PAVIS |
| | Random | 8.3 | 2.0 | 1.3 |
| Single-shot | Skeleton (NN) [58] | – | 21.1 | 28.6 |
| | Skeleton (SVM) [59] | – | 13.8 | 35.7 |
| | 3D RAM [26] | 47.5 | **30.1** | 41.3 |
| | Our method (CNN) | **66.8** | 25.4 | **43.0** |
| Multi-shot | Skeleton (NN) [58] | – | 39.3 | – |
| | Skeleton (SVM) [59] | – | 17.9 | – |
| | Energy Volume [70] | 14.2 | 25.7 | 18.9 |
| | 3D CNN+Avg Pooling [8] | 28.4 | 27.8 | 27.5 |
| | 4D RAM [26] | 55.6 | 45.3 | 43.0 |
| | Our method (CNN-LSTM+Avg Pooling ) | 75.5 | 45.7 | 50.1 |
| | Our method with attention from [88] | 75.9 | 46.4 | 50.6 |
| | Our method with RTA attention | **76.3** | **50.0** | **52.4** |

**GPT-4o**
The method achieving the highest Top-1 Accuracy for multi-shot person re-identification on the BIWI dataset is "Our method with RTA attention" with an accuracy of 50.0%. This outperforms the best single-shot method, "3D RAM (26)" which has an accuracy of 30.1% on the same dataset. **Correct**

**GT Answer:** The proposed method with RTA attention achieves the highest Top-1 Accuracy for multi-shot person re-identification on the BIWI dataset with a score of 50.0%. This is significantly higher than the best single-shot method on the same dataset, which is 3D RAM with a score of 30.1%.

R-L: 53.8    BERT: 85.5    LAVE: 9    L3S: 97.5

Figure H.3: **Example demonstrating the SPIQA CoT QA task**. Given a question and all the figures and tables, the AI system needs to identify which image is most helpful in answering the question and then provide an answer to the question. R-L: ROUGE-L, BERT: BERTScore, L3S: L3Score.

# I  Error Analysis

Fig. I.1 shows examples where all the models retrieve the correct figure (table or image) that helps answer the question. However, in many cases the models do not correctly answer the question. We observe that in the case of tables (represented as images), models have difficulty parsing and comprehending the information and making errors. This highlights room for further improvements for advanced systems in terms of comprehending table content represented as figures.

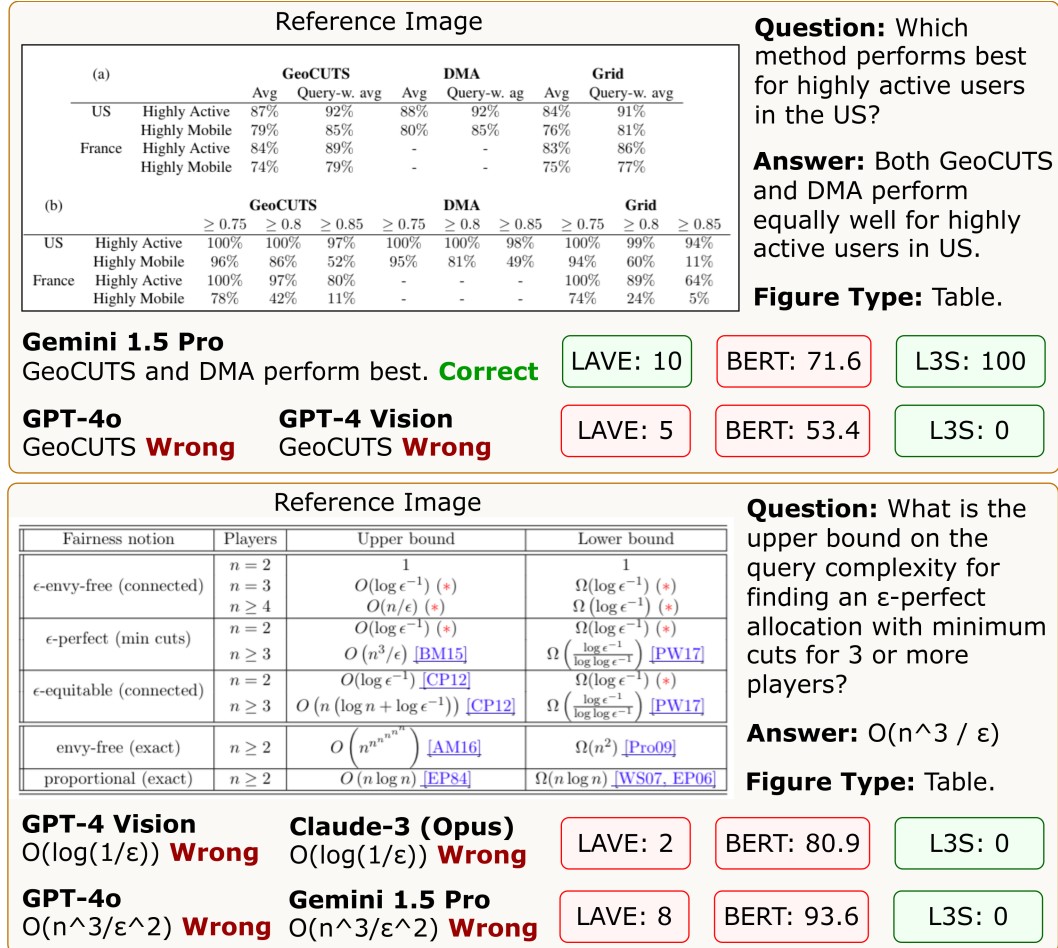

Figure I.1: **Error Analysis.** Examples of questions and answers where some models respond incorrectly. Models struggle to fully parse and understand the information in complex tables. The L3Score correctly indicates that the model responses do not capture the expected answer. Metrics colored in green denote correct evaluations, while those in red indicate incorrect scoring. BERT: BERTScore, L3S: L3Score.

