# OpenReview forum: "SPIQA: A Dataset for Multimodal Question Answering on Scientific Papers"
_NeurIPS.cc/2024/Datasets_and_Benchmarks_Track — NeurIPS 2024 Track Datasets and Benchmarks Poster_

### Official Review · Reviewer_m6KR · 2024-07-24
**Solid Work with Concerns about Novelty and the Circularity of Using LLMs to Generate and Assess QA Pairs**

**Rating:** 6
**Confidence:** 4
**Correctness:** Yes, claims are made correctly, and t…
**Clarity:** Yes, the paper is clearly written and…

**Review:**

The motivation behind the creation of SPIQA is well-articulated, addressing a clear gap in existing resources by focusing on visual information within scientific articles. The experiments conducted are appropriately executed, providing thorough and valuable insights into the current capabilities and limitations of multimodal models. The evaluation of 12 multimodal large language models is detailed, showcasing the dataset's utility and the models' performance.

Despite its strengths, the paper does raise some concerns. The novelty of the dataset may be questioned as it integrates elements from existing work, which could be seen as reducing its originality.
Another concern is the reliance on LLMs for both the generation and evaluation of the QA pairs. This approach might introduce biases and not accurately reflect the dataset's utility for human users. It would be beneficial to validate the questions generated by LLMs by asking human researchers to distinguish whether questions were generated by LLMs or humans. This additional validation step could help ensure the quality and relevance of the QA pairs.

**Strengths:**

see above

**Additional Feedback:**

no additional questions.

**Documentation:**

Yes.

**Ethics:**

no ethical concern

**Limitations:**

see above

**Opportunities For Improvement:**

see above

**Relation To Prior Work:**

Yes.

**Summary And Contributions:**

In this paper, the authors introduce a novel QA dataset designed to address the limitations of existing datasets, which focus solely on textual content in scientific papers. The dataset, containing 270K questions, leverages the capabilities of multimodal large language models (MLLMs) to understand and generate questions about figures, encompassing a wide range of visual data representations.
The paper evaluates 12 prominent foundational models on their ability to comprehend research articles and introduces a Chain-of-Thought (CoT) evaluation strategy to enhance model performance through in-context retrieval.

---

> ### Author Rebuttal · Authors · 2024-08-17
>
> We thank the reviewer for the thorough review and comments.
>
> **Q1. The novelty of the dataset may be questioned as it integrates elements from existing work.**
>
> As we note in Supplementary Section C (Table C), SPIQA addresses a significant gap in existing scientific QA benchmark datasets, and is the first large-scale free-form multimodal QA dataset on scientific research articles from various domains of computer science. We have designed a novel QA generation and eval pipeline by utilizing the ability of multimodal LLMs to understand text and single images but evaluate them on a more challenging multi-image task. Moreover, we develop three well-designed tasks for direct and retrieval-based CoT question answering to access baseline systems’ step-by-step fine-grained reasoning capabilities.  We further propose LLMLogScore (L3Score), a LLM-based evaluation metric for free-form QA, which addresses limitations of existing metrics by incorporating the confidence of LLMs to evaluate the quality of candidate answers using log-likelihood token probabilities. Our extensive experiments demonstrate the value of SPIQA and L3Score to assess the multimodal and long-context capabilities of several closed and open-source MLLMs. Our proposed dataset and code are made open-sourced for future research purposes.
>
> While we automatically generate the predominant portion of SPIQA, we also curate two additional test sets (test-B and test-C) with human-written QAs by filtering two existing text-only datasets, QASA and QASPER and significantly augment them by adding evidential figures for responses grounded in figures and tables to design a valuable multimodal testbed. Hence, we believe that SPIQA will add significant value to the community for accessing the capabilities of current large foundational models to understand multimodal scientific literature.
>
> **Q2. It would be beneficial to validate the questions generated by LLMs by asking human researchers to distinguish whether questions were generated by LLMs or humans.**
>
> Thank you for the suggestion. We tested whether annotators could tell if a question was written by a human or generated by a model. We gave them a mix of questions: 40% were written by humans and 60% were model generated, and asked them to choose one out of three options to decide whether the questions were generated by: human, machine or unsure. The results showed that:
>
> $\bullet$ 85% of the time, annotators were unsure whether the question was generated by a human or a machine.
>
> $\bullet$ 4% of the time, they mistakenly thought a human-generated question as a model-generated question.
>
> $\bullet$ Only 11% of the time they correctly identify a model-generated question.
>
> The results of the study concludes that LLM-generated questions have high quality and are almost indistinguishable from human-generated questions, which was further verified during the manual filtering process.
>
> **Q3. Circularity of Using LLMs to Generate and Assess QA Pairs.**
>
> This is a good question and relates to the quality of the dataset. Please refer to the general response where we detail three key aspects of our dataset generation strategy to ensure the quality of generated QAs, and how we have incorporated suggestions from reviewers including yourself to report additional results on inter-annotator agreement, and human evaluation performance on the task.

---

> > ### Comment · Reviewer_m6KR · 2024-08-21
> >
> > Thank you for the clarification. I will update my review score accordingly.

---

> > > ### Author Response · Authors · 2024-08-21
> > > **Thanks**
> > >
> > > We are glad we were able to address your concerns. Thanks for your suggestions and letting us know that you will update the score based on the rebuttal.

---

### Official Review · Reviewer_HCve · 2024-07-24
**Multimodal Question Answering Dataset on Scientific Papers**

**Rating:** 7
**Confidence:** 3
**Correctness:** Yes
**Clarity:** Yes

**Review:**

SPIQA is the first large-scale dataset specifically designed for multimodal question answering on scientific papers, addressing a critical gap in existing benchmarks. The dataset and tasks are highly relevant to multiple research areas, including natural language processing, computer vision, and scientific document understanding. The introduction of L3Score as a new evaluation metric shows thoughtful consideration of the limitations of existing metrics for free-form QA evaluation. See strengths and weaknesses below.

**Strengths:**

1. The inclusion of complex figures, tables, and full paper text in the QA tasks represents a significant advancement in assessing AI systems' capabilities to understand scientific literature comprehensively.

2. Focusing on computer science papers directly addresses challenges faced by researchers in interpreting and analyzing academic literature.

3. The introduction of L3Score as a new evaluation metric shows thoughtful consideration of the limitations of existing metrics for free-form QA evaluation.

4. The proposed tasks, especially the CoT QA, encourage the development of more advanced reasoning capabilities in AI systems.

**Additional Feedback:**

N/A

**Documentation:**

Yes

**Ethics:**

No ethical concerns.

**Limitations:**

Yes

**Opportunities For Improvement:**

1. In LLMLogScore (L3Score), consider still use top-n log probabilities as for a generalizable evaluation metric, then specify the n=5 in the following experimental section for API constraints.
2. Any human evaluation on test-A?
3. Additional analysis on the consistency between L3Score and other metrics.

**Relation To Prior Work:**

Yes

**Summary And Contributions:**

This paper introduces SPIQA (Scientific Paper Image Question Answering), a large-scale dataset for multimodal question answering on scientific papers. SPIQA is the first QA dataset designed to interpret complex figures and tables within scientific research articles across computer science domains.
It contains 270K questions divided into training, validation, and three evaluation splits, covering various types of plots, charts, tables, and diagrams.
The authors propose three tasks: direct QA with figures/tables, direct QA with full paper text, and chain-of-thought (CoT) QA with retrieval.
They introduce LLMLogScore (L3Score), a new evaluation metric for free-form QA that uses log-likelihood probabilities to assess answer quality.
Extensive experiments are conducted using 12 prominent multimodal large language models (MLLMs), including both closed and open-source models.

---

> ### Author Rebuttal · Authors · 2024-08-17
>
> We thank the reviewer for the thorough review and comments.
>
> **Q1. In LLMLogScore (L3Score), [...], then specify the n=5 in the following experimental section for API constraints.**
>
> Thanks for your suggestion. We will edit the paper to incorporate this.
>
> **Q2. Human evaluation on test-A.**
>
> Thanks for this suggestion. As noted  in our general response and in the response to **Q4** from reviewer $\textcolor{RoyalBlue}{yhyx}$, we perform human evaluation on a portion of test-A where the human evaluator is set with a task identical to that of the large multimodal models and has to identify the figure or table that’s most helpful in answering the question and write an answer.
>
> The human evaluator obtains a 94.1% score (top-1 accuracy) on retrieving the correct figure that helps answer the question, which is 8% higher than the best model performance. The METEOR, ROUGE-L, BERTScore F1, L3Score and CIDEr metrics are marginally better compared to the best models.  Kindly refer to the table in our response to **Q4** from reviewer $\textcolor{RoyalBlue}{yhyx}$ for the full set of scores. That said, we want to emphasize that it is difficult to obtain a strong upper bound on the SPIQA dataset since it would require human evaluators with expertise across the breadth of the areas covered in the dataset or have multiple experts evaluate different subsets which can be challenging.
>
> **Q3. Additional analysis on the consistency between L3Score and other metrics.**
>
> Thanks for the suggestion. We have computed the Spearman’s $\rho$ between METEOR, ROUGE-L, CIDEr, BERTScore F1 with L3Score for the responses of GPT-4o on test-A in the direct QA setting.
>
> | Comparison between metrics |  Spearman’s $\rho$ |
> | :----: | :--------: |
> | METEOR and L3Score | 0.71 |
> | ROUGE-L and L3Score | 0.72 |
> | CIDEr and L3Score | 0.69 |
> | BERTScore F1 and L3Score | 0.78 |
>
> In the table above, we notice a strong positive correlation between previous metrics with L3Score. However, as shown in Figure 4, and supplementary figures G.1-G.3 and H.1, L3Score analyzes the semantic meaning of candidate answers with respect to the ground-truth, and correctly scores the responses even if there is no token similarity. To support future research, we have made
> * the implementation of L3Score openly available (https://github.com/google/spiqa/blob/main/metrics/llmlogscore/llmlogscore.py) and
> * provide a starter code (https://github.com/google/spiqa?tab=readme-ov-file#-starter-code-snippet-for-l3score) for community use.

---

> > ### Comment · Area_Chair_cQWs · 2024-08-31
> >
> > hi reviewer HCve, thank you for your review! this is a gentle ping that the authors have responded to your review. does it address your concerns? if so, are you updating your score, if not, why? do you want to engage further in the discussion with the authors? appreciate the response!

---

### Official Review · Reviewer_yhyx · 2024-07-26
**Innovative and large scale dataset for scientific figure Q&A; uncertainty regarding validation**

**Rating:** 6
**Confidence:** 4
**Correctness:** My questions/concerns about correctne…

**Review:**

Pros:

- First large-scale dataset for scientific figure/table QA
- Introduces a new evaluation metric (LLMLogScore)
-Comprehensive evaluation of state-of-the-art models
- Informative error analysis

Cons:

- Lack of inter-annotator agreement metrics
- Uncertainty about test set quality and human performance
- Potential mismatch between question generation and evaluation conditions
- Limited improvement with full paper context

Quality
The overall quality of the work is good, but there. The dataset creation process is extensive and well-motivated, addressing a clear gap in the literature. The model evaluation is extensive.
However, the lack of inter-annotator agreement metrics and a human performance baseline raises questions about the reliability of the test set. The authors' manual filtering process lacks quantitative measures of its effectiveness. Some criteria for filtering seem to lack clear operationalization.
The introduction of LLMLogScore as a new evaluation metric is well-motivated and addresses limitations of existing metrics. However, more extensive validation of this metric would strengthen its contribution.

**Strengths:**

- SPIQA is the first large-scale dataset specifically designed for Q&A with scientific figures and tables. This addresses a gap in the literature.

- The dataset is impressively large.

- The paper introduces LLMLogScoreas as a new evaluation metric for free-form QA. It is well-motivated and addresses problems with some existing metrics.

- The study evaluates a wide range of state-of-the-art models.

- The fine-tuning experiments provide evidence of utility of the training set, despite lack of manual filtering or validation.

- The error analyses are informative regarding model performance.

- I did a manual evaluation of around 10 question/answer pairs in the dataset. The questions and answers are generally quite good, though see below for caveats.

**Additional Feedback:**

Thank you to the authors for an interesting and valuable dataset.

**Clarity:**

The paper is dense, and describes a lot of information, but I believe it was well written given the space constraints. The supplementary materials were helpful.

**Documentation:**

The dataset was easily accessible.

**Ethics:**

There are no ethical concerns.

**Limitations:**

The authors have only a brief discussion of limitations. There are no clear negative societal impacts of this work.

**Opportunities For Improvement:**

- The paper does not report any measure of agreement between multiple annotators for the test set validation process. Without this, it is difficult to assess the reliability of the human judgments used to validate the questions. Some of the criteria seem subjective, e.g. "The question is neither too trivial nor too specific to the figure or table."

- More generally, given that the question/answer pairs were generated by an LLM, the paper leaves some uncertainty about the quality of the test data. I manually inspected around ten questions from the dataset. Many of the question/answer pairs looked good, but some of them were very difficult to judge, and one of them was a clear error, reproduced below:

> {"question": "What is the purpose of the decoder in the 3D mesh prediction pipeline?", "answer": "The decoder is responsible for generating the final 3D mesh from the intermediate representations produced by the regression network.", "explanation": "The figure shows that the decoder takes the outputs of the regression network (Output 1 and Output 2) as input and produces the final 3D mesh. This suggests that the decoder performs some sort of post-processing or refinement on the intermediate representations to generate the final output.", "reference": "1812.00281v3-Figure12-1.png"}

- I have some uncertainty about the first test set validation criterion: "Answering the question would require a complete understanding of the figure and its importance in the paper." The LLM only gets to see the figure, its caption, and a passage that directly talks about the figure. How would it be able to generate questions that would require understanding the figure in the context of the whole paper?

- There is uncertainty about whether the generated questions are actually answerable using only the information provided in some task setups, particularly when full paper context is not available. The study does not establish a ceiling performance or human baseline for the direct QA with figures and tables task. Without this, it's hard to contextualize how well the models are actually performing relative to human capability.

- The relatively small performance improvement when models are given the full paper (compared to just figures/tables) is potentially concerning, though it could just be due to limitations of long-context models. It may potentially indicate problems with how the questions are formulated.

**Relation To Prior Work:**

The paper provides a clear and comprehensive discussion of prior work.

**Summary And Contributions:**

This paper introduces SPIQA (Scientific Paper Image Question Answering), a large-scale dataset designed to evaluate understanding of scientific figures and tables. The dataset comprises more than 270,000 question-answer pairs generated from more than 25,000 computer science papers, covering various subfields. The authors propose three task formulations: direct QA with figures and tables, direct QA with full paper text, and a chain-of-thought QA approach. The study evaluates a number of multimodal language models on SPIQA, including both open-source and closed-source systems. Additionally, the authors conduct fine-tuning experiments on two open-source models, demonstrating potential performance improvements. The paper aims to provide a benchmark for comprehension of scientific content in a multimodal context.

---

> ### Author Rebuttal · Authors · 2024-08-17
>
> ### **(Response 1/2)**
>
> We thank the reviewer for the thorough review and constructive comments.
>
> **Q1. The paper does not report any measure of agreement between multiple annotators for the test set validation process.**
>
> Thanks for the suggestion. To ensure the accuracy of our manual filtering process before scaling up to the entire evaluation set, we conducted multiple pilot annotation studies to determine the UI and annotator agreement. On the final UI, two experienced AI/ML researchers independently annotated 100 samples from the the evaluation set where the inter-annotator agreement score was 88%. Notably, the annotation task was designed to be straightforward, with annotators discussing and familiarizing themselves with the guidelines apriori to ensure consistency and making a simple binary choice between keeping or discarding each question.
>
> **Q2. More generally, given that the question/answer pairs were generated by an LLM, the paper leaves some uncertainty about the quality of the test data. [...], and one of them was a clear error, reproduced below: [...]**
>
> We acknowledge the concern raised by the reviewer and we address in our general response on how we maintain a good quality of generated questions, and have also incorporated your suggestions on reporting the inter-annotator agreement and human evaluation performance (with additional numbers in response to  your question in **Q4**) .
>
> With regard to this specific instance ("1812.00281v3-Figure12-1.png"), could you clarify more. We note that the answer is accurate and a correct response to the question.
>
>  We do observe that there is a minor mistake in the explanation, where the explanation should have said
> > “explanation”: The figure shows that the decoder takes the outputs of the regression network as input and produces the final 3D mesh.
>
> Notably, We did not ask the annotators to review and correct the explanation. We will make a note of this in the final manuscript. We have not used the explanations during the model training or evaluation; however we release the explanations for future research purposes. Moreover, to develop a robust and complete QA benchmark on scientific articles, we curate two additional test sets (test-B and test-C) consisting of human written questions and answers.
>
> **Q3. I have some uncertainty about the first test set validation criterion: [...] How would it be able to generate questions that would require understanding the figure in the context of the whole paper?**
>
> This is a fair criticism of the annotation guidelines in the way we have framed it. The annotators interpreted this as a slightly less stringent criterion, in the sense that they verified whether the question is relevant in the context of the paper, or is too specific to the figure.
>
> To illustrate this point, consider two examples. A question like "What is the blue block in the model diagram referring to?" would be discarded because it is too focused on a specific figure and doesn't require any deeper understanding of the paper. In contrast, a question like "What is the role of the cross-modal attention modules (indicated in blue) in the model architecture?" is more suitable because it requires comprehension of the paper's content. We will provide more explicit guidelines with additional examples in our final manuscript to ensure greater clarity.

---

> > ### Comment · Area_Chair_cQWs · 2024-08-31
> >
> > hi reviewer yhyx, thank you for your review! this is a gentle ping that the authors have responded to your review. does it address your concerns? if so, are you updating your score, if not, why? do you want to engage further in the discussion with the authors? appreciate the response!

---

> > > ### Comment · Reviewer_yhyx · 2024-08-31
> > >
> > > I would like to increase my score to a 7, but I'm not able to edit the review for some reason.

---

> > > > ### Author Response · Authors · 2024-08-31
> > > > **Thanks**
> > > >
> > > > Thanks for acknowledging our response and letting us know that you want to increase the score to a 7 but are unable to modify the review.
> > > >
> > > > We will add the human validation results, and  include limitations regarding the benchmark and correct the claim requiring understanding of the full paper and share the human upper bound on performance. We'll incorporate these and other suggestions in the final paper.
> > > >
> > > > Thanks again for your suggestions and helping improve this work!

---

> > ### Comment · Reviewer_yhyx · 2024-08-31
> > **Thank you for your response**
> >
> > My questions have been partially answered by the rebuttal. I think this is a good paper, and am increasing my score. I would recommend that the authors add the human validation results, and acknowledge limitations regarding the benchmark requiring understanding of the full paper and the human upper bound on performance.

---

> > ### Author Response · Authors · 2024-08-31
> > **Reviewer yhyx wants to increase score to 7 but is unable to edit the review**
> >
> > Dear ACs,
> >
> > Reviewer yhyx wants to increase the score to a 7 but is unable to edit the review. We hope you will consider this in the final decision.
> >
> > Thanks!

---

> > > ### Comment · Area_Chair_cQWs · 2024-09-01
> > >
> > > thank you reviewer and authors for engaging here.
> > > i acknowledge that reviewer yhyx wants to increase their score to 7 but were not able to do it through the ui 🤷‍♂️
> > > will take this into account

---

> ### Author Rebuttal · Authors · 2024-08-17
>
> ### **(Response 2/2)**
> **Q4. The study does not establish a ceiling performance or human baseline for the direct QA with figures and tables task. Without this, it's hard to contextualize how well the models are actually performing relative to human capability.**
>
> Thanks for the suggestion. One of the difficulties with getting a human upper bound is that the SPIQA dataset includes many different research domains from computer science, and hence, answering the questions requires broader expertise, and having a single human evaluator to cover the range is very challenging. Moreover, the free-form response can be quite varied depending on the evaluator’s knowledge of the field and understanding of the figures or tables. Based on the reviewer feedback, we obtained responses from a single AI/ML researcher on a fraction of test-A.  The human evaluator is given a task identical to that of the large multimodal models and has to identify the figure or table that’s most helpful in answering the question and then, write an answer. Their performance is reported below:
>
> | Method | Retrieval Acc. | METEOR | ROUGE-L | CIDEr | BERTScore F1 | L3Score |
> | :---- | :--------: | :-----: | :--------: | :--------: |  :--------: | :--------: |
> | Human Eval | 94.1 | 27.9 | 44.2 | 133.1 | 69.4 | 67.94 |
> | Gemini 1.5 Flash w/ CoT | 86.2 |  25.5 | 38.7  | 121.0 | 66.10 | 55.06 |
> | Gemini 1.5 Pro w/ CoT | 86.2 | 25.1 | 37.6 | 98.2 | 66.48 | 63.98 |
> | GPT-4 Vision w/ CoT  | 84.3 | 25.9 | 39.0 | 106.4 | 67.08  | 64.27  |
> | GPT-4o w. CoT | 86.2 | 27.0 | 43.2 | 130.0 | 68.10 | 66.14 |
>
> The human evaluator achieved a top-1 accuracy of 94.1% in retrieving the correct figure, outperforming the best model by about 8%. However, when it comes to answering the question, human performance marginally surpasses the best model scores. To establish a stronger human baseline, it would be necessary to  engage multiple human evaluators with diverse expertise which we will release in future versions.
>
> **Q5. The relatively small performance improvement when models are given the full paper (compared to just figures/tables) is potentially concerning, though it could just be due to limitations of long-context models. It may potentially indicate problems with how the questions are formulated.**
>
> We agree with the reviewer that the performance improvement using the full text from the papers is smaller than anticipated. As you astutely pointed out, this can be partly attributed to the models' limitations in processing long-context data. On average, every paper in the SPIQA dataset contains ~38000 tokens when including the full text, figures, tables and captions which poses a challenging task.
> As mentioned in our general response, we intentionally introduce an asymmetry between question generation and evaluation conditions to create a more challenging task for the models. While large multimodal models have made significant strides in single-image tasks, our goal is to push them further by testing their capabilities on multi-image long-context tasks with highly relevant content. By doing so, we aim to address the existing gap in their performance and better understand their strengths and weaknesses. Improving the question generation strategy to mitigate any remaining issues in quality is an important direction for future work.

---

### Official Review · Reviewer_oGaX · 2024-08-01
**This paper provides A Dataset of 270K questions for Multimodal Question Answering on Scientific Papers.**

**Rating:** 5
**Confidence:** 4
**Correctness:** yes
**Clarity:** yes

**Review:**

quality: The provided dataset is very useful for testing or enhancing the understanding of large multimodal models on CS related papers. But the 270K questions do not always have high quality and it is unclear whether the training on those potentially wrong data will encourage hallucinations.

clarity: Overall, the paper is very clear to me. It includes detailed statistics and experiments to support their claims. However, one of the biggest concerns is that the quality of the overall data is not guaranteed in this paper.

originality: Overclaim. This paper claims to provide the first large-scale QA dataset. However, such QA dataset has already existed as also noticed by the authors in Table C.1. The large-scale QA data is not guaranteed with high quality and only evaluation set was verified by humans.

significance: I admit the usefulness of the data since the fine-tuned models on this large-scale data enhance performance, but may also encourage hallucinations since quality is not guaranteed. Actually, I since the evaluation set undergoes human verification so it would be a good testbed for understanding the abilities of current LVMs.

**Strengths:**

See above.

**Additional Feedback:**

Can you provide an ablation study to show whether the performance gain comes from large-scale QA pair pretraining or simply unsupervised pertaining on those raw data?

**Documentation:**

yes

**Limitations:**

see other comments.

**Opportunities For Improvement:**

Could you also provide QA in other domains in CS? Since many publications were not on conferences.

**Relation To Prior Work:**

yes

**Summary And Contributions:**

This paper claims to provide the first large-scale QA dataset specifically designed to interpret complex figures and tables within the
context of scientific research articles across various domains of computer science. They also craft an information-seeking task involving multiple images that cover a wide variety of plots, charts, tables, schematic diagrams, and result visualizations. SPIQA comprises 270K questions divided into training, validation, and three different evaluation splits. They evaluate 12 prominent foundational models and propose a Chain-of-Thought (CoT) evaluation strategy with in-context retrieval that allows fine-grained, step-by-step assessment and improves model performance.

---

> ### Author Rebuttal · Authors · 2024-08-17
>
> We thank the reviewer for the thorough review and comments.
>
> **Q1. Originality: Overclaim. [...] However, such QA dataset has already existed as also noticed by the authors in Table C.1.**
>
> We politely want to clarify our claim. As we discuss in the supplementary section C, the existing large-scale scientific QA datasets (BioRead, BioMRC, emrQA) follow cloze-style queries, i.e. fill-in-the-blanks. In this setup, randomly selected named entities are masked out from sentences, and the task is to guess those missing words by reading the preceding passage. Hence, such cloze-form datasets fail to reflect real-world scenarios where the readers often have open-ended questions when reading a long scientific research paper. Existing datasets with free-form QAs on scientific literature (BioAsq, PubMedQA, QASPER, QASA) are extremely small scale, having a maximum of 5K training samples which is inadequate to fine-tune current neural networks to perform well on content from the scientific domain. Moreover, questions from such existing datasets only focus on the textual content of the papers, overlooking the wealth of information presented in figures and tables. SPIQA is the first large-scale free-form multimodal QA dataset on scientific research articles from various domains of computer science. We discuss various fine-grained differences of SPIQA with previous benchmarks in lines 67-73 in the supplementary.
>
> **Q2. Can you provide an ablation study to show whether the performance gain comes from large-scale QA pre-training or simply unsupervised pertaining using those raw data?**
>
> Thanks for the suggestion. We conduct an ablation study to investigate the impact of unsupervised pre-training using figure-caption pairs from the SPIQA training set and report the results in the table below. During pre-training, we ask the model to describe a given figure, using the caption as ground truth. This approach does not require the generated QAs.
>
> | Method | Mode of Eval (test-A) | METEOR | ROUGE-L | CIDEr | BERTScore F1 | L3Score |
> | :----: | :-------- | :-----: | :--------: | :--------: |  :--------: | :--------: |
> | LLaVA-1.5-7B | Zero-shot | 22.6 | 34.7 | 117.8 | 61.61 | 13.86 |
> | LLaVA-1.5-7B | Unsupervised Pre-training | 22.6 | 35.1 | 118.2 | 61.53 | 14.43 |
> | LLaVA-1.5-7B | Supervised Fine-tuning | 23.8 | 36.0 | 121.2 | 63.74 | 45.45 |
> | LLaVA-1.5-7B | Unsupervised Pre-training + Supervised Fine-tuning | 24.0 | 36.4 | 121.7 | 63.92 | 46.11 |
>
> As shown in the first two rows of our results, unsupervised pre-training yields only marginal improvement over zero-shot evaluation on test-A. In contrast, supervised fine-tuning using QAs significantly enhances performance, demonstrating the value of generated annotations for advanced question-answering. Furthermore, we found that combining figure-caption pre-training with QA-based fine-tuning leads to additional improvements, highlighting the potential benefits of large-scale QA data for developing specialized QA systems for scientific papers in the future.
>
> **Q3. The large-scale QA data is not guaranteed with high quality and only the evaluation set was verified by humans.**
>
> Our response to your suggestion in $\textbf{Q2}$ shows the utility of the SPIQA training set for developing advanced QA systems. We also detail three different steps we perform to ensure the quality of training data in the general response. To reiterate,
>
> 1. Our question generation is done in a simpler set up whereas the evaluations are done in a more complex set up.
>
> 2. We performed pilot studies with different powerful LLMs and prompts prior to scaling question generation.
>
> 3. We also do manual filtering of the test set. Manual filtering of questions is expensive, time-consuming, and largely relies on the expertise and knowledge of the annotators in the particular domain and the paper.  Since only around 11% of questions are discarded during filtering the evaluation set, which refers to the good quality of generated questions, we do not perform additional filtering on the train and validation splits.
>
> **Q4. Could you also provide QA in other domains in CS? Since many publications were not at conferences.**
>
> We have chosen to collect papers from conferences due to their open-source nature and permissive licenses, which allow us to utilize them for training AI models for research purposes. Our collection spans a diverse range of domains within computer science, including artificial intelligence/machine learning (AI/ML), natural language processing (NLP), computer vision (CV), information retrieval, databases, networking, data mining, audio, and signal processing (see supplementary material, lines 78-84). In contrast, papers published in journals such as IEEE Transactions and Springer typically require subscriptions for access, limiting our ability to collect, use, and share them. However, we note that some journals like TMLR, JMLR, and DMLR are open-source, and we can consider  incorporating papers from these sources into future versions of SPIQA.

---

> > ### Author Response · Authors · 2024-08-28
> > **Please verify the incorporated suggestions**
> >
> > Dear Reviewer oGaX,
> >
> > We see that your score deviates from the other reviewers. We have included your suggestions adding some additional evaluations, and we believe we have addressed all concerns. We would appreciate it if you are able to acknowledge our rebuttal and let us know if there are any further comments.
> >
> > Thanks!

---

> > > ### Comment · Area_Chair_cQWs · 2024-08-31
> > >
> > > hi reviewer oGaX, thank you for your review! this is a gentle ping that the authors have responded to your review. does it address your concerns? if so, are you updating your score, if not, why? do you want to engage further in the discussion with the authors? appreciate the response!

---

> > ### Author Response · Authors · 2024-09-04
> >
> > Hi Reviewer oGaX,
> >
> > We believe we have addressed all of the concerns raised in our rebuttal. Your score deviates from other reviewers, we would appreciate it if you are able to look at the responses and acknowledge the rebuttal.
> >
> > Thanks!

---

### Author Rebuttal · Authors · 2024-08-17

### **Thanks to all reviewers for their positive comments and valuable suggestions**

We thank the reviewers for their insightful and positive feedback. We are encouraged that they find our proposed SPIQA dataset to be well-motivated ($\textcolor{RoyalBlue}{yhyx}$), well-articulated ($\textcolor{RedOrange}{m6KR}$), highly useful ($\textcolor{brown}{oGaX}$), addressing a clear gap in the literature ($\textcolor{RoyalBlue}{yhyx}$, $\textcolor{green}{HCve}$, $\textcolor{RedOrange}{m6KR}$); our experiments to be extensive ($\textcolor{RoyalBlue}{yhyx}$, $\textcolor{green}{HCve}$), appropriately executed ($\textcolor{RedOrange}{m6KR}$), thorough and providing valuable insights into the current capabilities and limitations of multimodal models ($\textcolor{RedOrange}{m6KR}$); our proposed LLMLogScore, the novel free-form QA evaluation metric to be thoughtful and accurately addressing the limitations of existing metrics ($\textcolor{RoyalBlue}{yhyx}$, $\textcolor{green}{HCve}$). We appreciate that the reviewers also find the paper very clear ($\textcolor{brown}{oGaX}$, $\textcolor{RoyalBlue}{yhyx}$, $\textcolor{green}{HCve}$, $\textcolor{RedOrange}{m6KR}$), detailed ($\textcolor{brown}{oGaX}$, $\textcolor{RedOrange}{m6KR}$), easy-to-follow ($\textcolor{RedOrange}{m6KR}$), and well-written ($\textcolor{RoyalBlue}{yhyx}$). Moreover, we are thankful that all the reviewers agree that the proposed benchmark offers a significant and valuable advancement in assessing the capabilities of current large foundational models to understand multimodal scientific literature.

One common concern raised by reviewers ($\textcolor{brown}{oGaX}$, $\textcolor{RedOrange}{m6KR}$) has been the quality and relevance of LLM-generated questions; and the quality of the test set ($\textcolor{RoyalBlue}{yhyx}$). Based on the feedback and suggestions from reviewers, we conduct additional evaluations and report them in our rebuttal.

*  **Human evaluation performance** ($\textcolor{green}{HCve}$, $\textcolor{RoyalBlue}{yhyx}$). With the eval setup identical to that of the models, human performance on a subset of the test-A was 94.1% on retrieval (vs 86.2% for the best model performance), we have more human evaluation underway, we report other metrics below in the individual response to reviewers.

*  **Inter-annotator agreement for manual filtering** ($\textcolor{RoyalBlue}{yhyx}$). We measured agreement of two annotators on 100 samples from the test-A evaluation set and found them to have a high level of agreement of 88% on whether questions should be kept or discarded based on the guidelines.

*  **Distinguish questions generated by humans or LLM** ($\textcolor{RedOrange}{m6KR}$). We asked an annotator to distinguish questions as human or machine generated, where 40% were human generated and 60% model generated, the annotator rated 85% of the questions as indistinguishable (“unsure” from amongst human, machine and unsure).

$ $

We would also like to point out three things that we did in the question generation process to ensure the quality of generated QAs:

1. Firstly, our *asymmetric set up for question generation vs. evaluation.* Question generation is done in a very simple setting with a single image which current multimodal LLMs have shown remarkable progress on (e.g. LLaVA). This ensures that the generated questions and answers are accurate and high quality. However, during evaluation, we want the model to process all figures and tables from the paper together, ~11 on average, alongside captions and potentially full text, which tests the long context ability of the model making this a harder task.

2. Second, as described in lines 171-173, before scaling up the question generation process, we have conducted an *extensive pilot study with different powerful multimodal LLMs and different prompts*, which concluded that using Gemini 1.5 Pro in a controlled question generation setting with prompt shown in Figure E.1. results in accurate, good-quality and to-the-point questions.

3. Third, we perform *comprehensive manual filtering of the evaluation set* to ensure the quality. As described in lines 184-186, only 11% of QAs were discarded during evaluation set filtering, denoting the sufficient quality of questions generated by Gemini. Further, as suggested by reviewers ($\textcolor{RoyalBlue}{yhyx}$), we report the inter-annotator agreement score for evaluation set filtering on 100 questions and find the agreement to be at 88%, which denotes a small number of ambiguous and confusing questions.

Furthermore, as suggested by $\textcolor{brown}{oGaX}$, we perform **additional ablation study with unsupervised pre-training** using the SPIQA training data, and the results of these experiments reflect the effectiveness of our proposed benchmark (results table is in the individual response to $\textcolor{brown}{oGaX}$). We believe our novel and holistic QA annotation approach would open doors to generating high-quality QAs for many domains in the future, bypassing the need for expensive end-to-end manual annotation. We respond to the individual reviewer’s comments below and will incorporate all feedback in the final version. We will also add more highlights from our pilot experiments in the final supplementary manuscript.

We welcome further discussion as well as requests for additional clarifications.

---

> ### Author Response · Authors · 2024-08-23
> **We welcome further questions**
>
> Dear reviewers $\textcolor{brown}{oGaX}$, $\textcolor{RoyalBlue}{yhyx}$ and $\textcolor{green}{HCve}$,
>
> Thanks for taking the time to review our work and providing valuable feedback. Please see our rebuttal incorporating your comments. We look forward to your response and are happy to address further questions while there's an opportunity in the discussion period.
>
> Thanks!

---

### Decision · Program_Chairs · 2024-09-26

**Decision:**

Accept (Poster)

**Comment:**

Recommendation: accept

This paper introduces SPIQA (Scientific Paper Image Question Answering), a large-scale dataset for multimodal question answering on scientific papers. The dataset contains 270K questions across training, validation, and evaluation splits, covering various types of figures, tables, and diagrams from computer science research papers. The authors propose three QA tasks and introduce a new evaluation metric called LLMLogScore (L3Score). They conduct extensive experiments evaluating various multimodal large language models on the dataset.

Reviewers noticed the same strengths:

- SPIQA addresses a clear gap in existing datasets by focusing on visual information in scientific papers.
- large dataset covering various areas of computer science
- Thoughtful introduction of a new metric L3Score and experiments are well conducted

They also noticed similar weaknesses:

- There are concerns about the quality and validation of the automatically generated questions/answers.
- The improvement are marginal when models are fed the entire paper (and not just figures/tables/diagrams).

The authors provided a detailed rebuttal addressing many of these concerns:

- They reported inter-annotator agreement (88%) for the test set validation process.
- They conducted a human evaluation study showing the generated questions were largely indistinguishable from human-written ones.
- They provided human performance baselines on a subset of the test data.
- They clarified the novelty of SPIQA compared to existing datasets and detailed their QA generation strategy.

Overall, this a solid contribution and the quality of the discussion in the reviews reflect it.

I would encourage the authors to add the additional experiments they performed as part of the rebuttal to the paper, and to also frontload to the main body discussion about differences with other Scientific qa datasets.